# Multidimensional third-generation sequencing of modified DNA bases allows interrogation of complex biological systems

Serena S. David[1,4], Brendan A. Pacheco[1,4], Kensei Kishimoto[1], Sam Vantine[1], Kai Hu[1], Haibo Liu[1], Diana L. Davis[1], Hoang Tran[1], Benjamin F. Sallis[2,3], Levi Ali[1], Cole M. Haynes[1], Beth A. McCormick[2,3], Lihua Julie Zhu[1] & William A. Flavahan[1] ✉

DNA exists biologically as a highly dynamic macromolecular complex subject to myriad chemical modifications that alter its physiological interpretation, yet most sequencing technologies only measure Watson-Crick base pairing interactions. Third-generation sequencing technologies can directly detect novel and modified bases, yet the difficulty and cost of training these techniques for each novel base has so far limited this potential. Here, we present a method based on barcoded split-pool synthesis to generate reference standard oligonucleotides allowing novel base sequencing. Using novel base detection, we perform multidimensional sequencing to retrieve information, both physiologically stored and experimentally encoded, from DNA, allowing us to characterize the preferential replication of deleterious mitochondrial genome mutations, the infection dynamics of a host-pathogen model, and the effect of chemotherapy on cancer cell DNA at the single molecule level. The low cost and experimental simplicity of this method make this approach widely accessible to the research community, enabling complex experimental interrogation across the biological sciences.

Despite the complex panoply of physiological and pathological chemical modifications that occur in biological genomes[1–4], DNA sequencing approaches have relied on the Watson-Crick base-pairing properties of DNA to reveal the sequence of bases within a DNA molecule[5,6]. With the exception of a few methods that conditionally alter base-pairing properties based on chemical modifications of bases, such as bisulfite sequencing of 5-methylcytosine[7], this type of sequencing returns one dimension of information: the simple base sequence of the nucleic acid. While some approaches based on altered enzymatic activity at modified DNA bases (such as damID[8]) or antibody-dependent DNA immunoprecipitation[9] can characterize base modifications, these approaches can have base resolution or noise issues limiting their utility. Third-generation sequencing

technologies are capable of measuring beyond simple base-pairing interactions. Pacific Biosciences' SMRT sequencing is a sequencing-by-synthesis method that, unlike next-gen sequencing, can observe the kinetics of base addition, allowing slight differences in synthesis opposite a modified base to infer its modification[10]. Nanopore sequencing, including devices created by Oxford Nanopore Technologies, does not utilize sequencing-by-synthesis; instead the sequencer functions by passing a nucleic acid through an engineered pore and recording the current as the nucleic acid passes through; the observed current will depend on the five or six bases located within the pore as current values are measured[11]. Notably, a base modification will slightly alter this shape, and in many cases will cause a slightly different current than the unmodified base[12–15]. To

[1]Department of Molecular, Cell and Cancer Biology, University of Massachusetts Chan Medical School, Worcester, MA, USA. [2]Department of Microbiology, University of Massachusetts Chan Medical School, Worcester, MA, USA. [3]Program in Microbiome Dynamics, University of Massachusetts Chan Medical School, Worcester, MA, USA. [4]These authors contributed equally: Serena S. David, Brendan A. Pacheco. ✉e-mail: william.flavahan@umassmed.edu

recognize these current alterations, a reference current profile for the modified base must be experimentally generated; the dynamics of nucleic acid shape and pore current are too complex to derive from theoretical principles alone. The generation of a complete training library is expensive and technically challenging; as such the number of modified bases that can currently be identified by nanopore sequencing approaches is still limited. Some approaches have been implemented to bypass these technical challenges, including analysis of deviations from the reference current in either DNA with stochastically-incorporated base analogs such as BrdU[16], or in RNA with unknown modifications[17]. These approaches have proven quite successful yet still have limitations, including a requirement of a high alteration frequency and an inability for single-base resolution of novel/standard base heteropolymers. As such, these methods are impractical for bases which cannot be easily generated at great frequency.

Because of the challenges in the creation of these training libraries, the current modified DNA bases that can be sequenced by nanopore are limited to 5-methylcytosine and 5-hydroxymethylcytosine, 6-methyladenine, and BrdU/EdU, and accurate detection of these bases is often limited to certain base contexts or with poor base-pair level resolution. The potential bases which could be sequenced by nanopore may include many more bases encompassing base analogs, epigenetic modifications, and damaged or oxidized bases; if the training libraries for these bases could be reliably generated and sequenced. Indeed, several additional RNA bases can already be detected by Oxford Nanopore's Dorado tool, including 6-methyladenine, 5-methylcytosine, pseudouracil, and inosine, highlighting the potential that this device likely has to detect DNA modifications as well.

Here, we present a method involving barcoded split-pool synthesis (BSPS) to create training libraries for nanopore sequencing. Our approach allows for the procedural stochastic synthesis of every possible base combination of a desired length, and then uses the barcodes that were added as the base sequence was synthesized to determine the synthesized base string.

## Results

### Barcoded split-pool synthesis

In BSPS, we use procedural synthesis through repeated rounds of splitting, ligation, and pooling of a bead-bound common oligonucleotide to stochastically synthesize all possible combinations of a certain number of bases (the number of split reactions) of a certain $k$-mer length (the number of split-pool rounds performed) (Fig. 1a). Following each split, a single base (including standard and novel bases) is added to one end of the oligo while a corresponding barcode is added to the other. These barcodes allows for the identification of any novel bases in the synthesized $k$-mer, which during sequencing will likely be called as a standard base. The pool-mix-split step in between each of these ligation reactions ensures that some oligonucleotides get every single combination of bases (and corresponding barcodes) synthesized, the full combination of every base sequence is represented in the synthesized training library (Fig. 1b, c, see Methods for more details). This split-pool process is customizable in terms of the number of splits, the number of rounds performed, and the base(s) added in each split; it is also highly cost-effective (see Supplementary Table 1 for cost details and Supplementary Data 1 for oligonucleotide sequences).

We examined the feasibility of this approach to generate useful reference training libraries. We performed a five-round, five-base BSPS synthesis to generate a training library for deoxyinosine (dI, Fig. 1d). Inosine is a purine that can be formed from deamination of adenine, and will preferentially base-pair with cytosine, similar to guanine[18]. We performed one full synthesis reaction, using five pools (dA, dG, dC, dT, and dI) for five split-pool rounds (Fig. 2a, b). This synthesis yielded enough DNA for three library preparations, which were sequenced for 24 h each on a MinION nanopore sequencer. Combined, these sequencing runs yielded an average of 188-fold read coverage of the 3125 ($5^5$) possible five-mers (Fig. 2c). When these reads were sequenced, deoxyinosine residues were identified as deoxyguanines; however the barcoding strategy allowed for separation of reads where the basecalled dG was actually a dI, allowing the distinct dI current characterization (Fig. 2d). Separating these reads by the barcode

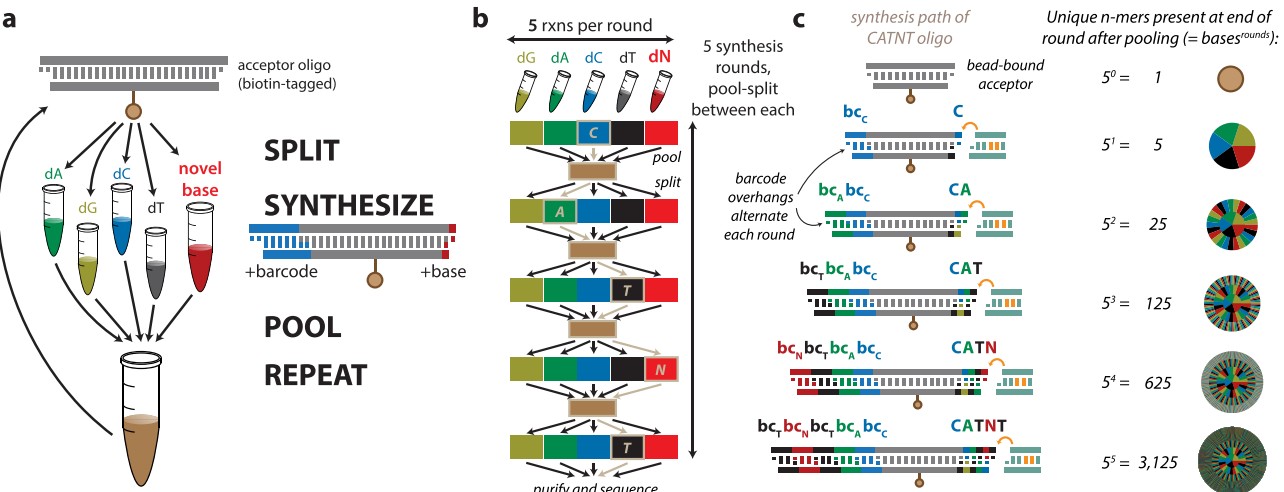

**Fig. 1 | Barcoded split-pool synthesis (BSPS) for the construction of novel base training libraries. a** Schematic displays split-pool approach. During each of $k$ synthesis rounds, the products of previous rounds (or an initial common 'acceptor' oligo) are split into one of $b$ split reactions, where a single base and a corresponding barcode are added. Following addition, these split reactions are pooled, mixed, and re-split. This process will create all possible $b^k$ combinations of $k$-mer base sequences, with barcodes corresponding to the encoded $k$-mers. **b** Split-pool strategy is outlined. In this example synthesis, five synthesis rounds of a five-base library, including the four standard base and a novel base N are performed. The specific path through the full oligo synthesis of an oligo encoding CATNT (with corresponding barcodes 3-2-4-5-4) is indicated through tan highlights. **c** The product of each split-pool step is highlighted. During each round, a single base is added to one of the growing oligo, while a corresponding barcode (bc) is added to the other end. The specific step-wise synthesis of the CATNT oligo is shown. Each synthesis reaction contains a mixture of all possible previous synthesis combinations; the number of distinct base/barcode sequences is shown at the right.

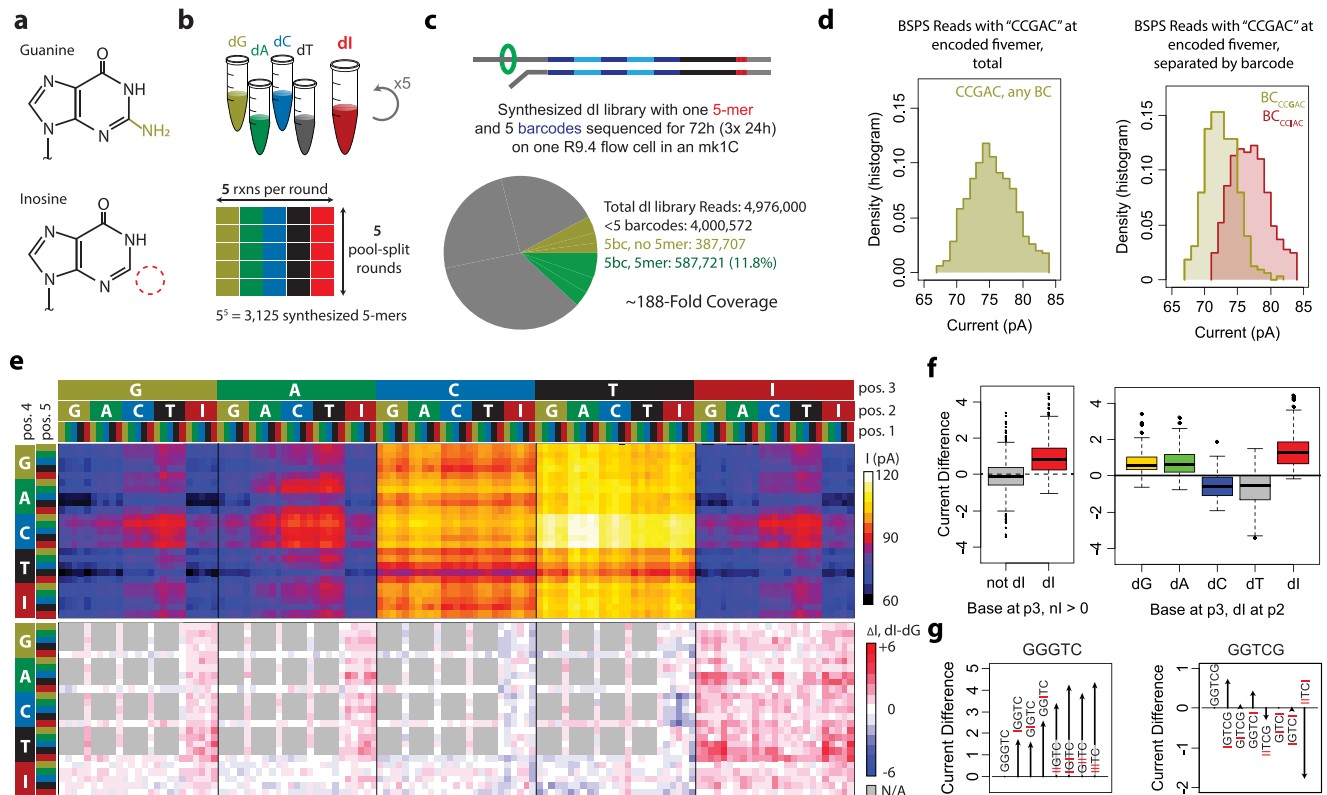

**Fig. 2 | Barcoded Split-Pool Synthesis of a deoxyinosine reference library. a** The chemical structures of guanine and inosine are indicated, with inosine's missing amine group highlighted by the red circle. **b** A five-round, five-split reaction was performed, with the four standard bases (dA, dG, dC, and dT) and dI. **c** The characteristics of the synthesized and sequenced library are presented. **d** Histograms depicts observed currents for all reads with a detected 'CCGAC' at the encoded fivemer (left), or these reads separated by barcode sequence (right) indicating either an encoded 'CCGAC' string (gold) or an encoded 'CCIAC' string (red). **e** Heatmap depicts full current values for all dI-containing five-mers. Top heatmap depicts absolute current value in increasing heat. Bottom depicts current difference of dI-containing fivemers to a reference five-mer with dG in place of each dI.

Red depicts a current increase of dI compared to dG, blue depicts a current decrease, and gray indicates five-mers with no dIs. **f** Box plots depict current differences of dI-containing five-mers with specific characteristics to the dG reference five-mers. Left, five-mers with dI in position 3 ($n = 625$) are compared to five-mers that contain dI but not at position 3 ($n = 1476$). Right, five-mers with dI at position 2 are separated by the base at position 3 ($n = 125$ per group). **g** Plots depict effects on current of specific G > I substitutions of two five-mers: GGGTC (left) and GGTCG (right). For box plots, center lines depict medians, box limits depict quartiles, whiskers depict 1.5x interquartile range, and points are outliers. Source data are provided as a source data file.

strings, we were able to characterize the entire deoxyinosine current landscape (Fig. 2e, Supplementary Data 2). The single-base resolution of our BSPS approach allowed us to identify trends in dI current alterations. For instance, a dG -> dI swap at position three consistently increases the current of the five-mer, while the effect of a dI substitution at position two depends on whether the base at position three is a purine or a pyrimidine (Fig. 2f). Additionally, in some contexts, such as GGGTC, dG -> dI swaps additively increase the current (Fig. 2g). However, in other contexts, such as GGTCG, the single- or double-substitution five-mers have minor and inconsistent effects on current, but the triple-swap (IITCI) has a significant decrease in current that could not be predicted from the single- and double-substitutions (Fig. 2g). These results show the utility of full coverage of the entire current landscape for novel base detection.

We next performed similar BSPS library syntheses on several base analogs (including deoxyuracil/dU and bromo-deoxyuracil/BrdU), modified bases (5-methylcytosine and 5-hydroxymethylcytosine), and damaged bases (O-6-methylguanine and abasic sites). In each case, these libraries allowed characterization of the nanopore current landscape of each base (see Methods, Supplementary Figs. 2,3, and Supplementary Table 2). Additionally, due to the ability of the BSPS libraries to reveal the entire current landscape of each base, we were able to calculate the statistical discrimination of the nanopore for single bases, and the ability to detect read- or region-level

incorporation of modified bases across multiple substitution or modification sites (Supplementary Figs. 4–7). These oligos were also used as the training input of a recurrent neural network (RNN), which was able to accurately classify withheld validation oligos (Supplementary Fig. 6a) and displayed high classification concordance with existing methods (Supplementary Fig. 6b, c).

### Tracking in vivo and in vitro nucleotide synthesis using two-dimensional sequencing

**Single molecule genetic and epigenetic drug sensitivity.** To examine the utility of this approach in analysis of genetically- or epigenetically-distinct cell populations, we investigated if cellular state and identity could simultaneously be distinguished by sequencing. We co-cultured two glioma cell lines with differential sensitivities to the tyrosine kinase inhibitor dasatinib[19], LN229 (resistant) and T98G (sensitive), in the same plate. These cells were allowed to grow for 24 h, then 10 µM dasatinib was added to the culture. BrdU was added to the culture 24 h later, and 24 h after that the cells were harvested and DNA was isolated and sequenced (Fig. 3a). DNA replication, as a surrogate of continued cell division, was determined via detection of BrdU incorporation (Fig. 3b). Reads were assigned to each cell line if they covered a cell-line specific SNP. By combining the sequence of genetically unique SNPs with BrdU incorporation, we were able to show that the sensitive T98G cells had a 90% drop in DNA replication in the dasatinib-

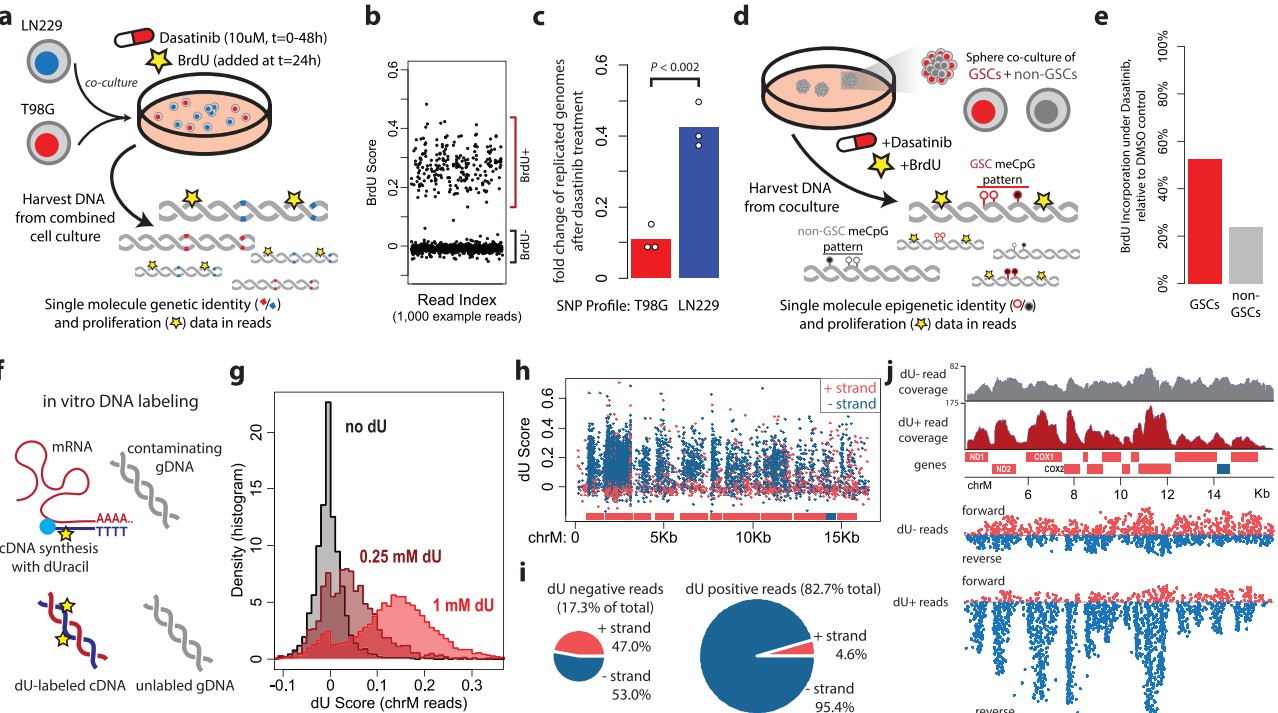

**Fig. 3 | In vivo and in vitro tracing of nucleotide synthesis with novel-base sequencing. a** Schematic depicts in vivo BrdU-labeling genetic single-molecule drug sensitivity experiment. Two glioma cell lines with differential dasatinib sensitivity (T98G – sensitive, LN229 – resistant) were cocultured in BrdU and dasatinib. Following isolation and sequencing of DNA, reads were assigned to either line by genetic single nucleotide polymorphisms (SNPs) unique to each cell line, and the response to drug was measured by BrdU incorporation. **b** Plot depicts BrdU detection in 1000 example reads, demonstrating separation of positive and negative reads. **c** Bar chart depicts single-molecule drug sensitivity as determined by BrdU incorporation in SNP-assigned reads. Significance calculated by two-sided *t* test, $P = 0.0018$. Points represent replicates and bar depicts average fold change. **d** Schematic depicts in vivo BrdU-labeling epigenetic single-molecule drug sensitivity experiment. BT142 glioma stem (GSC) and non-stem cells (non-GSCs) were co-cultured in dasatinib and BrdU (as in A); reads were assigned to stem or non-stem based on CpG methylation profiles. **e** Bar plot depicts single-molecule drug

sensitivity as determined by BrdU incorporation in meCpG-assigned reads. **f** Schematic depicts in vitro dU-labeling experiment. cDNA was synthesized with dUTP; synthesized cDNA should be labeled with dU while contaminating gDNA should not. **g** Histograms depict dU score (estimated error-corrected T > U replacement fraction) of reads with the indicated dU concentration during cDNA synthesis. **h** Plot depicts read centers by position along chrM (x-axis) and dU score (y-axis). Read color indicates strandedness; blue reads map to the light (-) strand, while salmon reads map to the heavy (+) strand. Mitochondrial genes are indicated by boxes along bottom of plot, with strand orientation indicated. **i** Pie charts depict fraction of reads that align to heavy or light chain, separated by presence of dU-labeling. **j** Plots depict coverage of the mitochondrial genome (excluding the rRNA loci), separated by strand and dU content. dU+ reads are more frequently found on the reverse strand and cluster around gene location, unlike dU- reads. Source data are provided as a source data file.

treated condition compared to the DMSO control, while the resistant LN229 cells had only a 60% drop (Fig. 3c), confirming the known individually-assayed sensitivity profiles[20]. These results demonstrate that sequencing reads from a known genetic source, in this case distinct cell lines, can be separated by genetic sequence to isolate a second modified base signal, in this case BrdU incorporation; thus a similar experiment based on genetic alterations beyond SNPs, such as CRISPR edits or heterogenous mutations, could likely be similarly disentangled.

We next sought to assign reads from a mixed population of cells distinguished by their epigenetic, rather than genetic, information. We cultured glioma stem and non-stem cells in a mixed culture, and treated with dasatinib and BrdU as above, with the hypothesis that the stem cells would be more resistant to therapy. Following DNA isolation and sequencing, reads were assigned to coming from either a stem or non-stem cell based on known $^{5m}$CpG methylation patterns (which were previously derived from reduced-representation bisulfite sequencing datasets generated from each pure population) (Fig. 3d). Following treatment, we saw about a 75% drop in DNA replication in non-stem cells compared to DMSO control, but only a 50% drop in the stem cells (Fig. 3e), demonstrating at the single-molecule level a drug resistance phenotype specific to the epigenetic identity of these cells, without the need for separation or labeling of the cells prior to sequencing.

**In vitro labeling of synthesized nucleotides.** We next attempted to use novel bases to label in vitro synthesized nucleotides. We performed a cDNA synthesis with deoxyuracil triphosphate (dUTP) spiked into the reaction, labelling our cDNA with dU. We analyzed reads that mapped to the mitochondrial chromosome, chrM, with the rationale that the small size and high copy number of chrM could result in significant genomic DNA contamination in the cDNA prep, even after DNase I treatment during RNA isolation[21,22]. Our hypothesis was that our synthesized cDNA would be distinguishable from contaminating DNA through the incorporated dU (Fig. 3f). Levels of detected dU within reads increased with increased dUTP in the reaction (Fig. 3g). Examination of the reads mapped to chrM revealed a strong association between dU incorporation and strand (Fig. 3h). While dU-negative reads did not display a strand bias, dU-positive reads were heavily biased towards aligning on the light (minus) strand (Fig. 3i). Most coding transcripts on the mitochondrial chromosome are located on the heavy (plus) strand; thus, this strand bias is consistent with first-strand synthesized cDNA. This was further confirmed by the location mapping on chrM. While the dU-negative reads are not biased to either strand and are evenly spread throughout the chromosome, the dU-positive reads are highly biased to the light strand and pile up over the location of the coding genes on the chromosome (Fig. 3j). Notably, we did observe frequent G- > A sequencing errors in certain contexts in

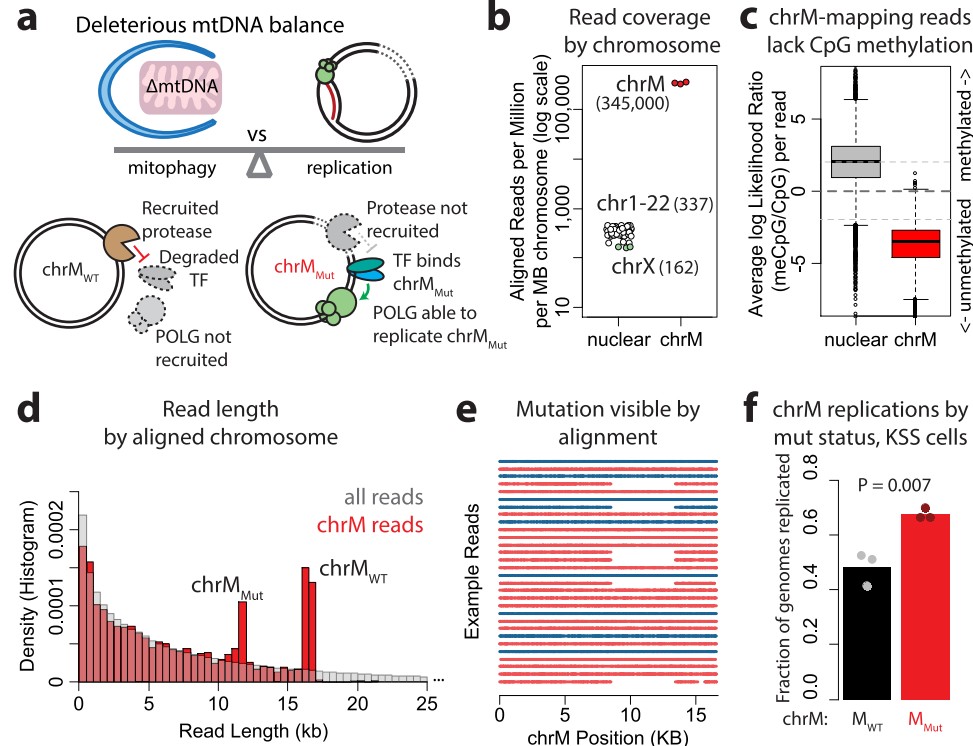

**Fig. 4 | Exploration of mitochondrial genome dynamics in human disease.**
**a** Schematic depicts paradigm of maintenance of deleterious heteroplasmy in human disease. Defective mitochondria are targeted for mitophagy more frequently, yet persist at high levels in the cell due to increased mutant chrM replication. On normal chrM, the protease LONP is associated with the mtDNA, and will degrade the protein ATF5 preventing it from binding to the DNA. On mutant chrM, the protease is not recruited, so ATF5 can bind and recruit POLG, leading to preferential replication of the mutated chromosome. **b** Plot depicts read coverage of each chromosome (in reads per million per MB chromosome); chrM has ~1000-fold higher aligned reads per MB than nuclear chromosomes. **c** Plot depicts detected $^{5m}$CpG on reads aligning to either chrM ($n = 3986$) or nuclear chromosomes ($n = 10{,}416$); plotted is the average log likelihood ratio of methylation across each read, with positive values being more likely methylated and negative values being more likely unmethylated. **d** Histogram depicts frequency of reads by length; total reads are plotted in gray, and reads aligning to chrM are plotted in red. **e** Example alignments of 20 example full-length chrM reads, colored by strand orientation. The ~5 KB deletion associated with KSS is clearly visible. **f** Replication of chrM genomes by mutation status, determined by BrdU incorporation. Reads representing chrM$_{mut}$ display elevated levels of replication. Points depict replicates (sequencing runs from independent cultures) while bars depict averages. Significance calculated by two-sided $t$ test, $P = 0.0069$. For box plots in this figure, center lines depict medians, box limits depict quartiles, whiskers depict 1.5x interquartile range, and points are outliers. Source data are provided as a source data file.

the dU-positive reads, which were predicted based on results from our full current characterization data (Supplementary Fig. 8).

## Characterization of complex biological systems using multidimensional sequencing

**Exploration of mitochondrial genome dynamics in human disease.**
We next attempted to link a disease-associated genetic variation with DNA replication alterations. Mitochondrial chromosome mutations are associated with numerous diseases; one specific deletion of ~5 kb is common in patients with Kearns-Sayre Syndrome (KSS)[23]. Previous work has shown that heteroplasmy of this deleterious mutation is maintained through alterations of mtDNA replication. While mitochondria with defective genomes are more frequently degraded by mitophagy, mutant genomes also replicate faster through use of the stress-associated replication response[24]. In healthy mitochondria, the protease LONP-1 is associated with the mtDNA and degrades the mitochondrial stress-associated protein ATF5. In dysfunctional mitochondria with the mutant genome, ATF5 is not degraded and can recruit the mtDNA replicative polymerase (POLG) to the mutant chrM, likely leading to increased levels of its replication (Fig. 4a). This increased replication can explain the persistence of deleterious genomes despite their increased likelihood of degradation. Importantly, replication levels have not been directly measured in this context.

We cultured a 143B cytoplasmic hybrid ("cybrid") cell line which is heteroplasmic, harboring both wildtype human mtDNAs as well as a population of mtDNA with the ~5 kb KSS-associated deletion[25], in BrdU for 24 h, and then isolated and sequenced DNA. Using an unmodified Oxford Nanopore rapid barcoding protocol, we obtained deep coverage of the mitochondrial chromosome without specific mtDNA enrichment; chrM had about 1000-fold higher coverage than that of the nuclear chromosomes (Fig. 4b). Notably, these mitochondrially-mapping reads lacked the characteristic CpG methylation seen in nuclear human DNA, confirming their mitochondrial origin through base modification chemistry as well as sequence (Fig. 4c). Examining the distribution of read lengths among reads that mapped to chrM, we observed two peaks corresponding to the linear length of the wild type mitochondrial chromosome ( ~ 16.5 kb), and that length minus the 5 kb deletion ( ~ 11.5 kb) (Fig. 4d). This suggested that many of our reads covered the entire mitochondrial genome, and that the mutant chromosome in these cells is a simple 5 kb deletion, and not a complex restructuring or concatenation of the mitochondrial genome. Next, we assigned chrM-aligning reads as either wildtype (chrM$_{WT}$) or mutant (chrM$_{Mut}$) based on the presence of the 5 kb deletion, or on co-segregating genetic variants for reads that did not cover the deleted region (Fig. 4e). Analysis of BrdU incorporation in these reads revealed that DNA replication rates were significantly higher in chrM$_{Mut}$ reads (Fig. 4f). Thus, multidimensional sequencing permits

direct measurement of the replication rate of mutant and wildtype mitochondrial genomes, confirming the established paradigm.

**Characterization of a human pathogen infection model.** We next sought to use multidimensional sequencing to characterize DNA derived from pathogen-infected human host cells, simultaneously profiling human and bacterial DNA without needing to separate human and bacterial cells prior to DNA isolation. We apically infected polarized cells from the colonic epithelial cell line T84 with *Shigella flexneri* type 2 A strain 2457T[26] for 24 h, then added BrdU to the media. Cells were harvested 1 h, 4 h, and 24 h after BrdU addition, and total DNA was sequenced (Fig. 5a). One-dimensional sequencing, looking at the proportion of reads that mapped to human or bacterial reference genomes, did not reveal infection dynamics (Fig. 5b). By contrast, two-dimensional sequencing using the alignment to either the human or bacterial genome and BrdU incorporation, accurately represented the biology of infection. Looking at DNA synthesis over time, we saw that the bacteria were dividing almost twice as fast as the human cells, and that the human reads in the infected condition replicated their DNA slightly less than uninfected controls at 24 h (Fig. 5c).

While we encoded the proliferation rate of cells (bacterial and human) into the reads through BrdU, we also reasoned that additional information was accessible in our reads. Specifically, we hypothesized that we could detect the source organism of a given read not just through whether we could align its genetic sequence to a known reference genome, but instead through the presence of epigenetic modifications unique to each organism. Human cells express DNMT enzymes which methylate cytosines ($^{5m}$C) in CpG motifs, yet CpG methylation is absent in the bacterial cells. *Shigella* express Dcm and Dam enzymes which add methylation to cytosines in CCWGG motifs and adenines ($^{6m}$A) in GATC motifs, respectively[27]. Our sequencing was able to detect these modifications in reads that aligned to each genome (Fig. 5d).

We examined the potential to isolate *Shigella* DNA through this unique base modification chemistry. Due to G/A sequencing errors, Dcm methylation was not useful for separation of unaligned (i.e. non genome-corrected) reads, however Dam methylation prediction remained highly accurate even without correction via a reference genome (Fig. 5e, f). From the sequencing reads that did not align to the human genome, we isolated reads that contained $^{6m}$A residues, yielding about 45,000 reads of a likely bacterial origin (Fig. 5g). Because these reads were identified chemically and not through their sequence alignment to a known reference, we were able to use the sequence of these reads to reconstruct the bacterial genome as if it had been unknown. Using flye[28], we assembled seven contigs comprising one genome (with three connected contigs of unsure orientation) and four plasmids (Fig. 5h). These contiguous sequences were highly consistent with the annotated genome and plasmid sequences known to be present in this strain of *Shigella*[29] (Fig. 5i). The contigs contained the complete and accurate sequences of the twenty one type 3 secretion system (T3SS) genes necessary for *Shigella* to infect human cells[30] (Fig. 5j). We also detected accurate sequences covering the *dam* and *dcm* genes, confirming the genetic competence of these cells for the associated methylation residues (Fig. 5k, l).

As a control for the utility of $^{6m}$A detection in genome assembly, 45,000 randomly selected non-human mapping reads were analyzed by flye, which did not predict a genome (Supplementary Fig. 9a). Analysis of the entire set of ~540,000 non-human reads (a set which contains the ~45,000 $^{m6}$A-positive reads) also failed to correctly assemble the bacterial genome (Supplementary Fig. 9b). In this analysis, several repetitive human sequences were incorrectly assembled as plasmids, and the assemblies of the genomic chromosome and the largest plasmid (which have many similar sequences) were intermixed in 86 highly-connected contigs. These results demonstrate the utility

and power of isolating reads of known bacterial origin via unique epigenetic modification.

Finally, we examined whether we could detect differences in the relative replication rates of each of the assembled plasmid sequences through detecting BrdU incorporation. Reads that aligned to the genome and each of the four plasmid contigs were analyzed for BrdU incorporation, revealing differences in their DNA replication rates (Fig. 5m, n). Notably, the pCP301 plasmid (contig_4), containing the T3SS genes, had a replication rate about half that of the genome during the 24 h of post-infection BrdU labeling. This may reflect regulatory mechanisms that govern the copy number of infection-critical yet energetically-demanding plasmids, pre- and post-infection, which merits further study. These results demonstrate that three-dimensional sequencing, measuring the sequence of nucleic acids, encoded DNA synthesis rates, and organism-specific epigenetic modifications, permits detailed interrogation of a complex pathogen infection model.

## Multidimensional sequencing permits single-molecule interrogation of chemotherapy response

We finally sought to characterize multiple encoded variables within a single read, looking at the paradigm of chemoresistance in cancer. Glioblastoma tumors are clinically treated with the alkylating agent temozolomide (TMZ)[31]. TMZ causes several alkylated base modifications, such as N7-methylguanine and N3-methyladenine, but the most clinically relevant is O-6-methylguanine (O6mG)[32]. O6mG will mispair with thymine rather than cytosine, yet mismatch repair (MMR) pathways will incorrectly view the paired thymine, rather than the O6mG, as the error, and are incapable of repairing the damage. If the O6mG is not repaired, this mismatch leads to futile cycling of the MMR response and cell death. O6mG is directly repaired through the enzyme O6-methylguanine-DNA methyltransferase (MGMT), which can covalently remove the methylation. The gene encoding this protein is frequently silenced in glioblastoma by promoter CpG methylation, rendering these tumors clinically sensitive to TMZ[31] (Fig. 6a).

We cultured T98G cells, a GBM cell line with about 40% *MGMT* promoter methylation, in TMZ for 48 h, with BrdU added at T = 24 h. These cells were harvested and the native promoters of *MGMT* were isolated for nanopore sequencing using a Cas9-based approach (Fig. 6b). Analysis of the $^{5m}$C current signatures at the *MGMT* promoter resolved reads as methylated or unmethylated (Fig. 6c). We hypothesized that temozolomide treatment may induce a selective advantage conferred to *MGMT*$^{me}$ cells that removed the inactivating methylation, as they would be able to re-express MGMT and repair the DNA damage. To test this hypothesis, we examined reads of methylated promoters for 5-hydroxymethyl-cytosine ($^{5hm}$C), a modification catalyzed by the TET enzymes as the first step in the removal of DNA methylation. We did not detect any signature of $^{5hm}$C in *MGMT*$^{me}$ reads. (Fig. 6d). $^{5hm}$C was detected in unmethylated reads however, suggesting active maintenance of the unmethylated promoter state in those cells. We next detected BrdU in the *MGMT* promoter reads with determined methylation states, and reconstructed drug sensitivity based on the epigenetic state of the *MGMT* promoter at the single-molecule level. As expected, reads with methylation at the *MGMT* promoter were from cells more sensitive to TMZ and displayed lower frequencies of BrdU incorporation than unmethylated reads (Fig. 6e), thus confirming the link between epigenetic state and drug sensitivity at the single-molecule level.

We next examined the reads for direct signatures of O6mG residues. Using values from our synthesized training libraries, we identified a read with a likely O6mG current value (Fig. 6f). This read also had a hairpin oligonucleotide, which we had added during the adapter ligation step of the sequencing library generation, successfully ligated, covalently linking both strands of DNA and returning sequencing data for both strands. BrdU signatures were detected in only a single strand

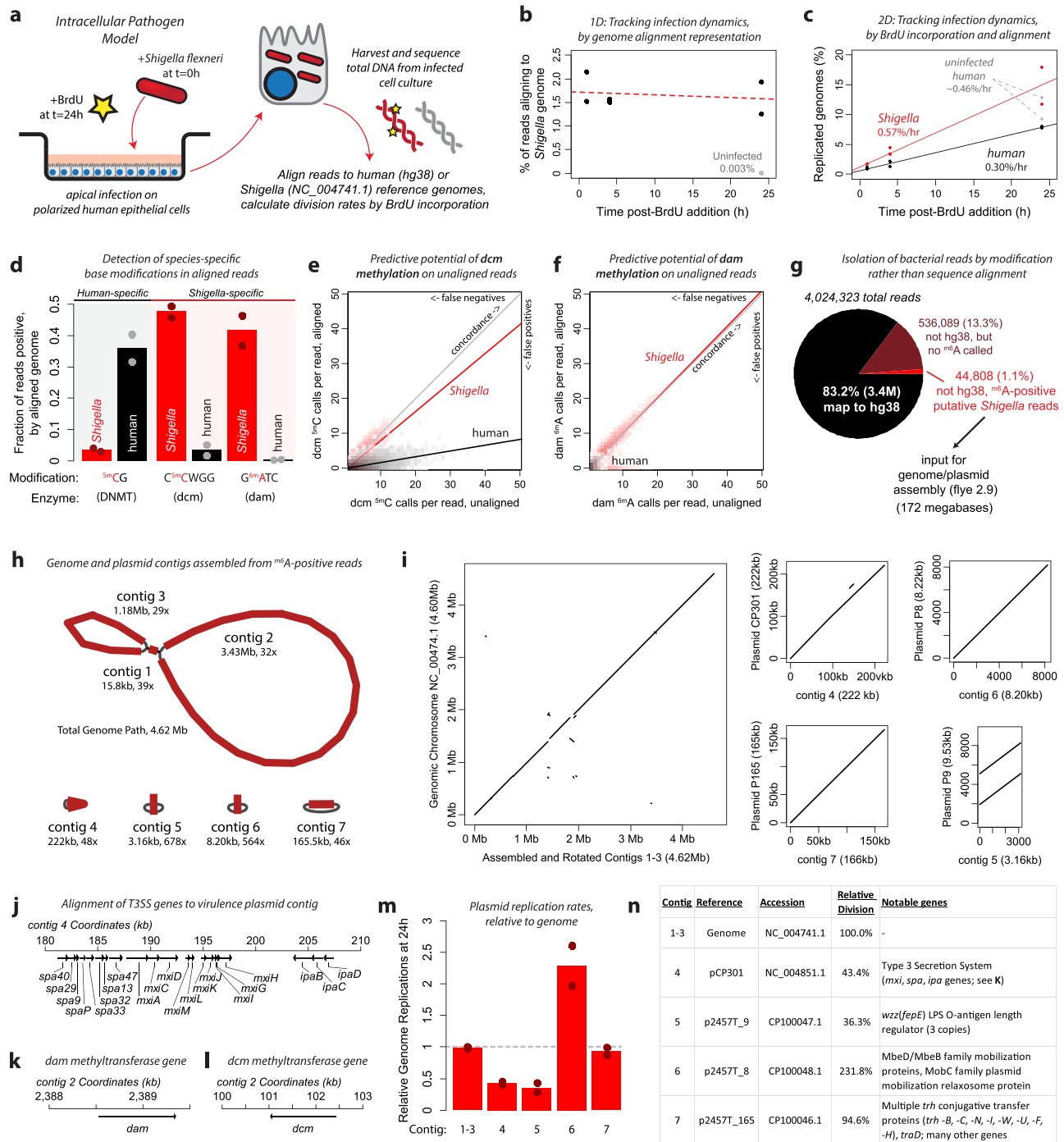

**Fig. 5 | Complex characterization of a human pathogen infection model.**
**a** Schematic depicts bacterial infection model; polarized colonic epithelial cells were infected apically with *Shigella flexneri* for 24 h, then BrdU was added to the culture media. Total DNA was isolated and sequenced. Two infections were performed and sequenced independently, points in panels b, c, d, and m represent results from each infection. **b** Plot depicts one-dimensional sequencing analysis, showing the proportion of reads from each timepoint that aligned to the *Shigella* reference genome. **c** Plot depicts two-dimensional sequencing analysis, showing the genome divisions detected by BrdU+ reads separated by alignment to either the human or *Shigella* genomes. **d** Bar graph depicts levels of the indicated base modifications in reads that aligned to either the human or *Shigella* genomes. **e**, **f** Plots depict concordance between dcm (E) and dam (F) methylation with reads either unaligned (x-axis) or aligned to the respective reference genome (y-axis). Alignment permits correction of sequencing errors (e.g. a C5mCAGG called sequence aligning to a C5mCGGG reference sequence). **g** Strategy of modification-

based bacterial read isolation. Reads from all infected conditions across both timepoint replicates were aggregated, and 6mA was detected in non-hg38 aligning reads; positive reads were used as input for genome assembly in flye. **h** Contigs generated by flye are depicted, along with sizes and read coverage. Three contigs (1-3) comprised the main bacterial genome and contigs 4-7 are circular plasmids of the indicated size. **i** Concordance between assembled contigs and reference genomes are depicted by pairwise alignment graphs. **j** The alignment of the twenty one type III secretion system genes to assembled contig 4 are depicted. The locus architecture is identical to the reference. The alignment of the reference sequence for the *dam* (**k**) and *dcm* (**l**) methyltransferases to the assembled contig 2 are depicted. **m** Relative genome divisions (calculated from BrdU read positivity) are depicted for the genomic assembly as well as each of the assembled plasmids. **n** Table depicts the corresponding reference, relative division rate, and notable genes of each plasmid. Source data are provided as a source data file.

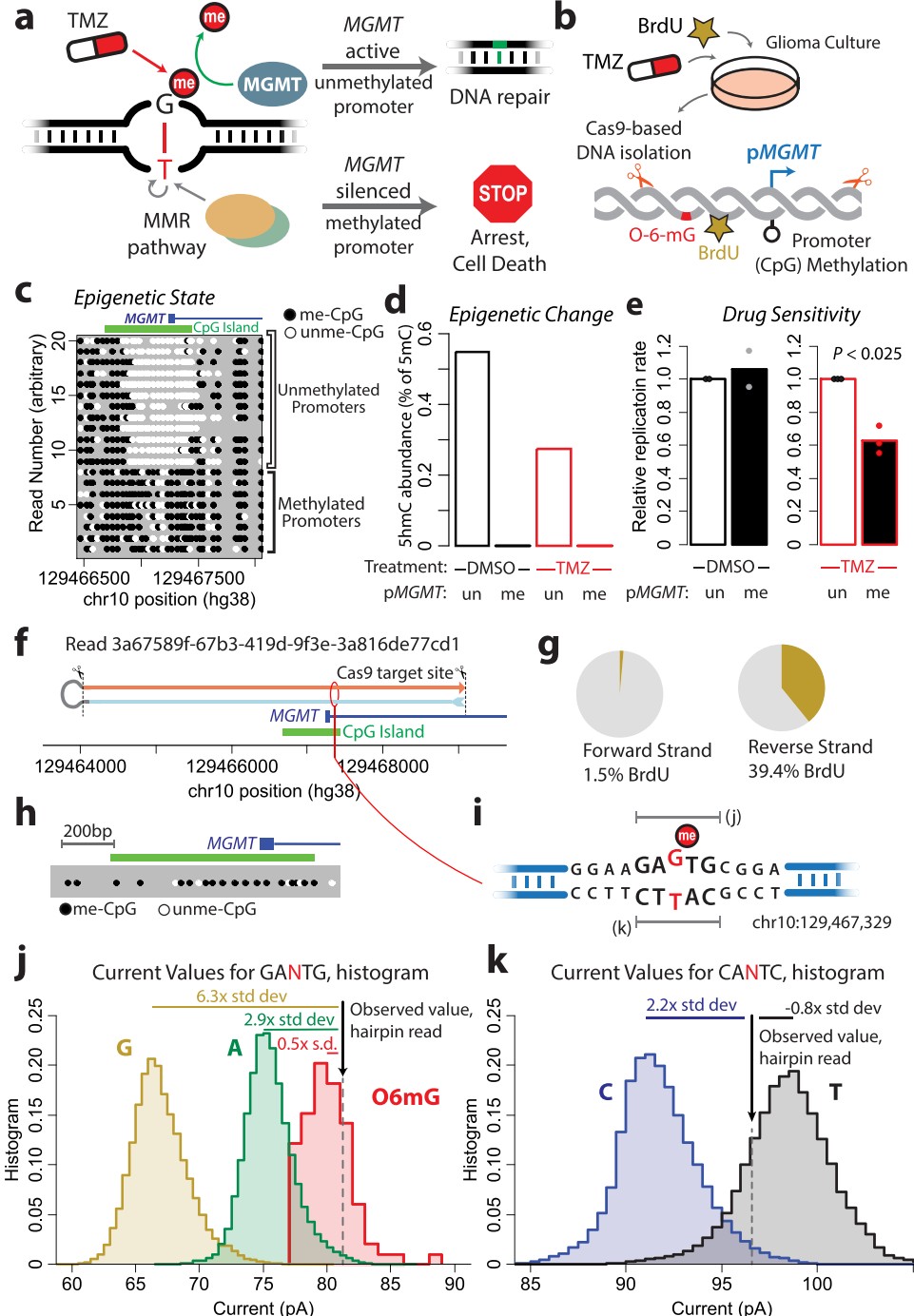

**Fig. 6 | Multidimensional sequencing reveals the dynamics of chemotherapy response in Glioblastoma. a** Schematic depicts mechanism of temozolomide (TMZ) action. *O*-6-methylation of guanine residues results in mispairing to thymine; this lesion is recognized by mismatch repair pathways but not repairable through MMR. Instead, the enzyme MGMT can directly remove the aberrant methylation. **b** Schematic depicts experimental setup; glioma cells were cultured in TMZ and BrdU, then the DNA from the MGMT promoter was isolated via Cas9, and sequenced. **c** Plot depicts methylation status at the CpG island overlapping the *MGMT* promoter. Reads are represented as rows, black circles depict methylated CpGs and white circles represent unmethylated CpGs. Methylation status of the *MGMT* promoter is clearly distinguishable. **d** Bar plot depicts detected 5-hydroxymethyl-cytosine at reads overlapping the *MGMT* promoter, separated by drug treatment and read methylation status. **e** Bar plots depict relative DNA replication (detected by BrdU incorporation) in pr*MGMT* reads, separated by methylation status, derived from both cells treated with 20 µM TMZ or DMSO. Drug conditions were sequenced separately and are normalized to each unmethylated promoter BrdU frequency. Points depict replicates treated and sequenced independently, significance was calculated for TMZ-treated cells by two-sided *t* test, *P* = 0.024. **f** Schematic depicts a single read from above experiment, which is characterized further in G-K. **g** The read in F had detected BrdU on the reverse strand ( ~ 40% T > BrdU replacement) and not on the forward strand. **h** Read displayed promoter methylation at *MGMT*. **i** At a single fivemer within this read, we detected a signature of O6mG (see **j**), with the opposite strand fivemer displaying a current consistent with a mispaired dT (see **k**). **(j)** Histograms depict current values for reference fivemers (GAGTG, gold; GAATG, green) are depicted, compared to detected current of read (black line). Also indicated is training library current data for O6mG (red). **k** Histograms depict current values for reference sequences at CACTC (blue) and CATTC (green), with read current indicated with black line. Source data are provided as a source data file.

of this read; consistent with a read that had replicated during the experimental period (Fig. 6g). Additionally, methylation analysis confirmed that this read originated from a methylated promoter (Fig. 6h). At one specific position within this read, we detected a current inconsistent with the expected G reference (Fig. 6i). This current value was six standard deviations above the reference mean for a dG, and almost 3 standard deviations above the dA reference. However, it was closely aligned (within one s.d.) with our O6mG data (Fig. 6j). At the corresponding position on the opposite strand, the current was more consistent with a dT than a dC, suggesting this read contains an O6mG mispaired to a dT (Fig. 6k). Finally, we also examined reads for signatures of abasic sites generated by our training library, as a marker of possible nucleotide excision repair, but no such currents were detected. Thus, in this experiment we extracted seven dimensions of information: (1) genetic sequence to confirm the read originated from the *MGMT* promoter, (2) epigenetic status of the promoter through CpG methylation, (3) active changes in the epigenetic status through CpG hydroxymethylation, (4) the proliferation status of the cell the read originated from through BrdU incorporation, (5) the direct effects of the alkylating agent through O6mG, (6) the indirect effects of the alkylating agent through the opposite-strand mispairing with dT, and (7) any signature of nucleotide excision repair through detection of abasic sites.

## Discussion

Here, we present a method for synthesizing training oligo libraries for novel or modified bases in any series of base contexts and *k*-mer length, which can generate reference currents for direct nanopore sequencing. We also demonstrate the power of this approach through several example experimental paradigms across the biomedical sciences. Our approach has numerous advantages over previous work; namely our complete coverage of novel/standard base heteropolymers, the inexpensive cost, and the ability to generate libraries for bases that are rare or challenging to work with. The low cost and relative ease of this approach also provides future-proofing; as new flow cells with different pore chemistry are commercially released, new training libraries can be synthesized and sequence to characterize the novel base current profiles of the new pores; the flexible nature of the protocol allows easy adaptation for new pore design (Supplementary Fig. 10). In this work, we synthesized mostly 5-mers, but the approach can be easily modified to generate reference *k*-mers of any relevant length. Once synthesized and sequenced, the data generated can also be shared with the research community to allow novel base sequencing by any lab with a nanopore sequencer. The complete coverage of heteropolymers also allows for detailed interrogation of the sensitivity and specificity of the nanopore itself for each modified base. Additionally, while calling a single base as modified or standard is statistically simple, these data can serve as the input for training more complicated machine learning-based approaches designed to combine the modification likelihood of multiple nearby bases for the detection of biological phenomena such as replication forks or modification-based protein-DNA footprinting.

Our exploration of multidimensional sequencing paradigms highlight the potential impact retrieving information from DNA beyond the simple base sequence. Using sequencing of total isolated DNA, we can analyze rare or hard to isolate cell populations, such as cancer stem cells within a tumor, without requiring prior enrichment of cells that may not have a consistent or convenient surface marker. We can track the synthesis of DNA during in vitro reactions, or link genetic or epigenetic state to DNA synthesis in vivo. Our analysis of *Shigella* infection dynamics also highlights the power of multidimensional sequencing; by isolating reads of bacterial origin via base modification chemistry rather than alignment to a known reference, we were able to isolate enough bacterial-origin reads to accurately assemble a complete reference genome including not just the genomic chromosome,

but also several plasmids encoding genes critical for the clinical pathogenesis of this organism. While this strain of *Shigella* is a well-characterized laboratory strain, our results demonstrate that we could have assembled an accurate genome of a pathogen with an unknown genome or plasmid complement, based purely on the detected base modification chemistry. Finally, in our examination of the characteristics of TMZ treatment, we demonstrate the potential for high-level multidimensional sequencing; encoding and retrieving multiple dimensions of information from a single sequencing read and fully characterizing the dynamics of this chemotherapeutic at the single-molecule level. These approaches demonstrate the power and utility of multidimensional sequencing; the detection of modified bases allows not just the study of that base itself, it also enables experiments deeply characterizing complex biological systems by using an organism's genome as a biological 'notepad' to encode experimental information.

Given the potential experimental applications of multidimensional sequencing presented here, and the increasing adaptation of third-generation sequencing approaches such as nanopore, this method represents an important tool for the research community. The generation and characterization of the training libraries in this work are immediately useful tools for other labs already performing nanopore sequencing; additionally the methods and analysis here should be useful to research across the biological sciences. Further expansions to this technology, to allow for characterization of modified bases in future nanopore technologies, or to characterize RNA in addition to DNA, could increase this potential even further.

## Methods

### Barcoded split-pool synthesis
We start with a single short ( ~ 60 bp) double-stranded acceptor oligonucleotide containing an internal desthiobiotin and bound to magnetic streptavidin beads. This oligo contains a four-base 5′ overhang, for barcode addition, and a one-base 3′ overhang, for donor oligo ligation (Supplementary Fig. 1a). These beads are then divided into five or more 'split' reactions, during which a single base in each 'split' is added to the 3′ end of the acceptor, while a corresponding barcode is added to the 5′ end (Fig. 1a). The barcodes alternate between 'even' and 'odd' overhangs to prevent multiple barcode addition (Supplementary Fig. 1b). The donor oligo is 'self-cleaving'; it contains an asymmetric type IIS restriction enzyme site that results in cleavage of the entire ligated oligo except for the added base(s), regenerating the one-base 3′ overhang (Supplementary Fig. 1a).

Forward and reverse oligonucleotides were designed and ordered from IDT (sequences are presented in Supplementary Data 1). Complementary oligos were annealed in Tris-HCl containing LiCl, and then phosphorylated using T4 PNK (NEB). Each split initial ligation reaction was prepared containing the following annealed oligos: desthiobiotin-labeled universal acceptor (1 μM), base-specific barcodes (4 μM) and corresponding barcodes containing 'odd' round overhangs (1 μM), and then ligated with Blunt/TA Ligase Master Mix (NEB). Following initial ligation, oligos were bound to streptavidin magnetic beads (Pierce) and unincorporated oligos were washed off. Bead-bound oligos were treated with 5 units of BciVI restriction enzyme (NEB) to cleave the donor oligo and regenerate the donor overhang, and then the beads were washed. Each split reaction was combined into a single pool, which was mixed and re-split into new tubes for the next ligation synthesis round. For subsequent rounds, barcode overhangs alternated between 'even' and 'odd' to prevent addition of multiple barcodes. Donor overhangs were adjusted based on the prevalence of each base addition in the previous round; for the initial round, the 3′ overhang of the acceptor was an adenine, so all donors had 3′ thymine overhangs. For a synthesis round following a mixture of bases, an even mixture of each base was present in the donor mixture. For a synthesis round following the addition of (methyl/hydroxymethyl)CpG bases, all donors had 3′ cytosine overhangs (Supplementary Fig. 1c).

The number of splits, number of rounds, and base composition of each split was experiment-specific, as described in the relevant section of the manuscript. Following the final synthesis round, each split reaction was treated with 5 units of SapI restriction enzyme (NEB), to uncover a distinct overhang for 3' hairpin oligo addition. Splits reactions were combined into a single pool, and 1.5 μM of hairpin oligos specific to the 5' and 3' overhangs were added and ligated with Blunt/TA Ligase Master Mix (NEB). Following hairpin ligation, beads were washed, and then synthesized oligos were recovered from the beads through washes with biotin to displace the desthiobiotinylated oligos from the streptavidin. Recovered oligos were purified via Ampure XP beads (Beckman Coulter), as per manufacturer's protocol. Oligos were then treated with Exonuclease VIII (NEB) to remove any oligos without both 5' and 3' hairpins, and purified via Ampure XP beads. Finally, oligos were treated with 5 units of AhdI restriction enzyme (NEB), to remove the 5' hairpin and uncover a 3' A overhang suitable for ligation to sequencing adaptors, and the final oligo pool was purified via Ampure XP beads and quantified via Qubit high sensitivity DNA assay (Thermo Fisher) and analyzed for size distribution via Tape Station (Agilent).

An example protocol is included as Supplementary Note 1. Details of synthesized and sequences BSPS libraries are presented in Supplementary Table 2.

## Nanopore sequencing of BSPS oligos

200 fmol of recovered oligo pools were prepared via the Ligation Sequencing Kit V10 (Oxford Nanopore Technologies), as per the manufacturer's protocol starting from the 'Adapter ligation and clean-up' step (thus omitting the end repair and A-tailing steps) and sequenced on a R9.4.1 flow cell and a Nanopore Minion MK1C. Each library preparation was sequenced for 24 h; subsequent library preparations from the same synthesized oligo pool was prepared and sequenced on the same or subsequent flow cells to obtain sufficient read coverage as necessary.

## BSPS data analysis

Sequencing data was basecalled during sequencing using Guppy (integrated into MinKNOW running on the Mk1C). Basecalled sequenced reads (fastq files) were aligned to all possible fully-synthesized oligos (R commands to generate this reference fasta file are included as supplemental data) via minimap2[33] with secondary alignments disabled. Reads that fully mapped to the reference were isolated using the 'view' command in samtools using the '-e rlen' parameter[34]. Raw signal values of full-length reads were isolated from the total fast5 files using ont_fast5_api (Oxford Nanopore Technologies). Reads (*.fastq), alignments (*.bam) and raw sequencing information (*.fast5) were processed to event current values using the 'eventalign' program of nanopolish[13–15], using both the --scale-events and --samples parameters. Current values corresponding to the encoded fivemers were isolated from processed eventalign files, and used to generate mean current, standard deviation, and read coverage values for each synthesized k-mer.

## Modified basecalling

Event currents from experimental reads were obtained via nanopolish eventalign as described above. Currents for potential modified bases were compared to reference canonical and modified base currents generated from BSPS via z-score. Modified base scores (e.g. Fig. 3b, g, h) are calculated as the fraction of potential modified base sites with z-scores matching the modified reference current, with a noise correction performed by subtraction of fraction of impossible modified base sites (e.g. called dG/dA/dC modifications in a dT/BrdU score). Example R commands for the computational processing of sequenced reference libraries and calculation of modified base incorporation are included in Supplementary Note 2.

## Thymidine analogue BSPS library synthesis

We performed BSPS synthesis preparations for the thymidine analogs dU and BrdU, again performing a five-round, five-base synthesis for each base[35,36] (Supplementary Fig. 2a). Sequencing of the dU library resulted in just over four million reads, with about a 16% final success rate, yielding on average 213-fold read coverage of the 3,125 five-mers (Supplementary Fig. 2b, top). Sequencing of the BrdU library resulted in over three million reads, but with a 22% success rate, resulting in a final average coverage of 234 reads per five-mer (Supplementary Fig. 2b, bottom). These datasets again resulted in the complete characterization of the base's current behavior in each case (Supplementary Fig. 2c, e). For deoxy-Uracil, the current is consistently elevated compared to dT. A dT -> dU substitution at position 3 seems to result in the largest increase, however there is also a large increase due to substitutions in other positions (Supplementary Fig. 2d, left). Indeed, a dT -> dU substitution results in an increased current at any of the five positions, and multiple substitutions increase the current further (Supplementary Fig. 2d, right). BrdU, however, displays a more complex current behavior (Supplementary Fig. 2e). A dT -> BrdU substitution at position 3 results in a significant increase in current similar to dU, although unlike dU substitutions at other positions have inconsistent effects (Supplementary Fig. 2f, left). For instance, a BrdU at position 4 is associated with a decrease in current if the base at position 2 is not BrdU and the base at position 3 is either a dG, a dA, or a dC. With a dT at position 3 the current effects are inconsistent, and a second BrdU at position 2 was enough to counteract the decreased current caused by BrdU at position 4, resulting in increased current even with a dG, dA or dC at position 3 (Supplementary Fig. 2f right). Full extracted current values are presented in Supplementary Data 3 (BrdU) and Supplementary Data 4 (dU).

## Motif addition BSPS library synthesis

In the previous experiments, our 'self-cleaving' donor oligos were designed to add a single base; but the design allows for the addition of any number of bases upstream of the restriction enzyme cleavage site. To demonstrate this, we generated a library containing CpG dinucleotides, with the cytosines being either unmodified, methylated, or hydroxymethylated[37] (Supplementary Fig. 3a). The donor oligos were designed to add two nucleotides at once; we designed a BSPS strategy with two donor oligo rounds; either adding CpG / meCpG / hmCpG, or a standard A/G/C/T single base (and one of seven corresponding barcodes) and performed a total of four rounds to create 108 different seven-mers (Supplementary Fig. 3b). Sequencing this library displayed the utility of having mixed heteropolymers of the cytosine modifications (Supplementary Fig. 3c). The change in current from converting a cytosine to a methyl-cytosine, or methyl-cytosine to hydroxymethyl-cytosine, depended on the modification status of the other nearby cytosine; in some cases opposite effects on current were observed from the same modification, depending on the nearby base context (Supplementary Fig. 3d).

## Post-synthesis BSPS library modification

The BSPS method also allows for post-synthesis modification to alter bases; one such post-synthesis base is abasic (AB, also apurinic/apyrimidinic or AP) sites. AB sites form in DNA either spontaneously following hydrolysis of purines, or as the first step in base excision repair following DNA damage such as cytidine deamination[38,39] (Supplementary Fig. 3e). AB sites, which completely lack the chemical structures involved in base pairing, cannot base pair and as such cannot be either sequenced by sequencing-by-synthesis methods, or directly synthesized by BSPS. However, as AB sites can be reliably generated enzymatically from deoxyuracil, we were able to generate an abasic site training library by first synthesizing a deoxyuracil training library, and then treating it post-synthesis with the uracil deglycosylase (UDG) enzyme[40] (Supplementary Fig. 3f). Initial tests suggested that five-mers

with multiple AB sites in close proximity were unstable; as such we generated the library with a single AB site, located in the third position of a five-mer. As may be expected, abasic sites display a consistently elevated current (Supplementary Fig. 3g). Indeed, every single sequenced abasic-containing five-mer displayed a current higher than every other reference current with any other base that we have sequenced in position 3 (Supplementary Fig. 3h). Further analysis of this data revealed that there was a strong sequence-related bias in coverage (Supplementary Fig. 3g, bottom). This appeared to be due to an imbalance in the conversion of reads from dU; by removing the final quality control filter, we were able to observe two distinct current peaks for the five-barcode reads. For high-coverage five-mers, like CT-abasic-GA, almost two thirds of the five-barcode reads had a current consistent with an abasic site, while the remaining third had a current consistent with an unconverted dU (Supplementary Fig. 3i). For a low coverage five-mer, such as GG-abasic-TA, less than 5% of reads had a current consistent with the abasic site, and the remaining 95% of reads were consistent with the unconverted dU (Supplementary Fig. 3j). Whether this observation is due to sequence-specific efficiency of the UDG enzyme, or oligonucleotide stability of the base context around the abasic site is unknown (perhaps certain base contexts around an AB site are more likely to result in a single-strand break); in either case the impact on the ability to derive reference currents for abasic sites is low, given the vast current difference between these sites and other nucleotide combinations. Full extracted current data for abasic sites are presented in Supplementary Data 5.

## Statistical analysis of modified base detection

Sequenced BSPS libraries allow for complete characterization of the Nanopore's ability to detect modified bases (Supplementary Fig. 4a). In order to calculate the discrimination potential of the pore, we calculated Receiver-Operator Characteristic (ROC) values for each 5-mer with a BrdU in position 3 from our BrdU BSPS library. Briefly, for each of the 2,101 BrdU-containing k-mers ($5^5$ total 5-mers minus the $4^5$ 5-mers containing just A,C,T,G, or $3125 - 1024 = 2101$) we set cutoff thresholds at the observed current value for each reference 5-mer (e.g. for CABBG, the reference 5-mer is CATTG), and calculated the true positive and false positive rate at each threshold. These were plotted to generate the ROC curves, and the area under the curve (AUC) for each ROC was calculated by summing the trapezoidal area for each threshold and the one before it (Supplementary Fig. 4b, c).

In order to calculate "test characteristics", that is, the ability to distinguish whether a read (or region within a read) has modified base incorporation, we performed Monte Carlo simulations to create and analyze simulated reads based on the known standards (Supplementary Fig. 4d). First, we took the subset of 5-mers that had a ROC AUC higher than 0.7, which corresponded to 48 reference 5-mers, and designated these as "callable" 5-mers. Given that there are 1024 total reference 5-mers ($= 4^5$), we thus expect a "callable" 5-mer to occur roughly once every $1024/48 \approx 21$ bases. For the Monte Carlo simulations, we then created 10,000 reads with $n$ randomly-selected callable 5-mers each. For each read, we created a dT read where current values for the reference 5-mer for each of the callable 5-mers was randomly selected from the appropriate BSPS data, and a BrdU read, where the current values were randomly selected from the BSPS data corresponding to the BrdU-containing 5-mer (the number of samples of each 5-mer that we could sample from is listed in Supplemental Data S3). For instance, if an $n = 3$ read had randomly selected the callable 5-mers CATCC, GATCG, and CTTCC, then the dT read had current values randomly selected from each of those three BSPS oligos while the BrdU read had current values corresponding to CABCC, GABCG, and CBBCC. For each read, the difference between the observed value and the current threshold (the current at which the single-base ROC distance from diagonal was maximal, calculated for each k-mer) was summed and used as the read score. The ROC was

then calculated over the 10,000 simulated reads, with the thresholds set at each dT read and the fraction of true and false positives calculated at each threshold (Supplementary Fig. 4e, f).

Next, we performed the same Monte Carlo analysis, but this time including a BrdU replacement fraction value ($f$). In this analysis, the current value for each dT read was calculated identically as above, but the current value for the BrdU reads was either randomly selected from the BrdU BSPS values (with probability $f$) or the reference dT BSPS values (with probability $1-f$). Thus, if the BrdU replacement value was $f = 0.3$, then on average 30% of the callable 5-mers in the BrdU reads corresponded to BrdU current values while the remainder were consistent with the dT current values. ROCs and AUCs were calculated for various values of $n$ and $f$, as shown (Supplementary Fig. 4g–i).

In order to verify the validity of the modified base calculation approach in real-world data, we cultured mouse 3T3 fibroblasts and human 293 T cells separately, with 100 μM BrdU added to the human cells alone (Supplementary Fig. 4j). DNA was isolated from each population, mixed, prepared via a rapid barcoding kit, and sequenced on an R9.4 flow cell. Reads were aligned to either mm10 or hg38, and only reads that aligned uniquely to a single genome were considered for analysis. Reads were analyzed for BrdU content either via DNAscent[16] v3.1.2 or using BSPS data as described above. For reads aligning to the mouse genome, both DNAscent and BSPS-based analysis called the overwhelming majority (99.6%) of reads as BrdU negative; as these cells were not grown in BrdU, this also serves as a measure of the specificity of each approach (Supplementary Fig. 4k). For reads aligning to the human genome, DNAscent and BSPS-based analysis agreed on 99.84% of reads, either both declaring those reads BrdU positive or both declaring the read BrdU negative (Supplementary Fig. 4l). The true positivity fraction was not determined, but these data suggest the sensitivity of each measure is similar. In addition, we determined the amount of callable 5-mers in each of these real reads; the majority of reads had $n$ values far above required for accurate BrdU incorporation detection (Supplementary Fig. 4m). We also calculated the relationship between callable 5-mer count and read length, with the reads containing a callable 5-mer approximately once every 20.8 bases, very close to the calculated value (Supplementary Fig. 4n).

We also used our known standards from the BSPS synthesis to evaluate the ability of the published standard DNAscent to accurately detect single BrdU incorporations. We analyzed our oligos with the reference CTTAT and BrdU incorporations at several points in that 5-mer. In each of the BSPS oligos, the 5-mer is the only incorporated BrdU location; thus the number of BrdUs per read can only be between one and three. Consistent with the demonstrated poor sensitivity of the nanopore at single-base BrdU incorporations (see Supplementary Fig. 4b, c), DNAscent did not accurately detect the incorporated BrdUs (Supplementary Fig 5a, b). These data, combined with the human/mouse analysis and previous publications utilizing DNAscent, confirm that single-base detection of BrdU is challenging but an approach based on combining several k-mers can allow highly accurate detection of read- or region-level incorporation. Indeed, analysis of currents for these 5-mers from BSPS sequenced data confirms frequent overlap in current between BrdU-containing and BrdU-free 5-mers, confirming the statistical difficulty of distinguishing BrdU incorporation at the single base level (Supplementary Fig. 5c).

## BrdU recurrent neural network data preparation

Raw multi-fast5 files from the BrdU BSPS library were converted into pod5 files using pod5 (v0.3.15). Basecalling was performed with the pod5 files as input using Dorado (v0.9.1), which output fastq files. Meanwhile, the raw multi-fast5 files were converted to individual single-fast5 files using ont-fast5-api (v4.1.3). Corresponding read information from the fastq files was written into each single-fast5 file using the preprocess annotate_raw_with_fastqs subcommand of Tombo (v1.5.1). Resquiggle was performed using the resquiggle

subcommand of Tombo (v1.5.1). At this step, Tombo performed genome mapping, signal normalization, event detection sequence, to signal assignment, and resolve skipped bases. The fastq reads were aligned to the reference sequences (theoretic sequences derived by assuming all split-pool synthesis processes are as expected). For any multi-mapping read, only the one with the best mapping score was kept. Any alignments with mapping quality <20 and alignment length <270 bases were removed. For the surviving reads, the normalized mean values (denoted as fm), standard deviation (fd), the number of current signals (fl), base of the 5mer were extracted. For each event, each base was one-hot encoded into a 4-vector feature, with 1 indicating the real base type, whereas 0 means otherwise. Therefore, a 7-vector feature was used to describe an event, denoted as $xi = [fm, fd, fl, fA, fC, fG, fT]$. The ground truth base type at the middle position of a 5mer was determined by the alignment process as described above, and one-hot encoded into a 5-vector feature denoted as $yi = [fA, fBrdU, fC, fG, fT]$, which was used as the truth label.

Similarly, two multi-fast5 files (one for each species) from the nanopore sequencing data for the mouse 3T3 (BrdU-free) and human 293 T (BrdU for 24-hour culture) cell lines (described above) were processed to get current information (normalized mean value, standard deviation, the number of current signals, and base) of 5mers with thymine in the middle position. The only differences are as follows: (1) the reads were aligned to the reference genomes of the corresponding species. (2) Any best-scored alignments with mapping quality <20 and alignment length <500 bases were removed. Similarly, the features of the 5-mers were prepared for both the mouse and human data, while the truth labels were only prepared for the mouse data.

## BrdU recurrent neural network training

The synthetic 5-mer data and 0.01% of the randomly sampled 5-mers from BrdU negative DNA (reads derived from 3T3 cells grown in BrdU free media) were used to train (80%) and test (20%) a recurrent neural network (RNN) after shuffling multiple times. The RNN was composed of three layers of bidirectional gated recurrent units (GRUs) and one dense layer. It took normalized mean values, standard deviations, lengths and one-hot encoded bases of each 5mer as input. The hyperparameters were tuned based on models' performance on the test data. In the favorite model, each GRU contained 128 neurons and batch size was set to 128. The model was trained for 10 epochs. The model's performance on synthetic testing data and 100,000 randomly selected mouse testing 5-mers are shown in Supplementary Fig. 6a.

The resulting model was used to predict BrdU bases incorporated into reads from DNA harvested from 293 T cells grown in BrdU, with validation 3T3 sequencing data as control. Human reads with much higher percentage of thymine basecalled as BrdU than the highest percentage of thymine basecalled as BrdU across all the mouse reads were considered as BrdU+ reads.

## BrdU recurrent neural network data and code availability

The code used to preprocess the synthetic data, the human and mouse cell line data, and model training are available at GitHub: https://github.com/haibol2016/NanoporeBrdUCaller. Training and testing data are available in the SRA accession included with this manuscript.

## Modified basecalling of non-BrdU bases

In addition to BrdU, we performed single-base ROC AUC calculations for abasic sites, deoxyuracil, and deoxyinosine (Supplementary Fig. 7). The single-base discrimination of dU and dI was highly similar to BrdU. In contrast, the current difference caused by an abasic site was so dramatic that the single-base sensitivity and specificity of every abasic 5-mer was above 97%. Unlike BrdU, there are no published alternate methods for nanopore-based detection of dU, dI, or abasic sites to compare with the BSPS libraries.

## In vitro drug sensitivity assays

For genetic (SNP) experiments, LN229 and T98G cells (both ATCC) were co-plated at a 1:1 ratio in the same well of a 6-well plate in DMEM with 10% FBS. Cells were treated with 10 μM dasatinib (Aobious) or DMSO 24 h post-plating. At 48 h post-plating, 100 μM BrdU (Thermo) was added to the media. Cells were harvested at 72 h post-plating and DNA was isolated via Quick-DNA kit (Zymo). DNA was prepared for sequencing using a Rapid Barcoding Kit (Oxford Nanopore) and sequenced on an R 9.4.1 flow cell for 24 h.

Basecalling was performed with Guppy, and basecalled reads were aligned to the human genome (hg38) using minimap2. Raw current values were obtained using nanopolish, as described above, and BrdU positivity of reads were determined as described for dU except with the calculations performed with the appropriate BrdU reference values. Values of the fraction of BrdU+ reads were converted to replicated genomes using the following formula: (% BrdU+) / (1 - %BrdU+). That is, in a population with 50% BrdU+ positive reads, there had been 100% divided genomes as on average each individual genome had replicated once to create one BrdU- strand and one BrdU+ strand. SNP array data was obtained for both cell lines from the Cancer Cell Line Encyclopedia (CCLE)[20], and a list of informative SNPs was curated; only SNPs which were homozygous for opposite alleles in each cell line were considered. Reads that covered these SNPs and the allele represented was obtained using the 'phase-reads' command in nanopolish. Reads were assigned to a cell type if they both covered at least 2 SNPs and had at least 80% agreement of all SNPs with the cell line's reference; other reads were discarded as uninformative. Experiment was repeated with three individual co-cultures of the same cell lines, result from each replicate is depicted as points in Fig. 3c. Statistical difference was analyzed by two-sided t-test.

For epigenetic (meCpG) experiments, BT142 cells were grown either in stem cell media (Neurobasal (Thermo) supplemented with 1X B27 (Thermo), 0.5X N-2 (Thermo), 20 ng/ml FGF2 (Peprotech), 20 ng/ml EGF (R&D Systems), and 1.5X Glutamax (Thermo) and penicillin/streptomycin) or differentiation media (DMEM with 10% FBS and P/S) for one week prior to experimental conditions. Stem and non-stem states were confirmed via qPCR for *OLIG2* and *GFAP*. Stem and non-stem cells were co-cultured at 1:3 stem:non-stem ratio in stem cell media supplemented with 2% FBS. 24 h post-plating, 10 μM dasatinib (Aobious) or DMSO was added to the media. 48 h post-plating, 100 μM BrdU was added to the media. Cells were harvested at 72 h post-plating and DNA was isolated via Quick-DNA kit (Zymo). DNA was prepared for sequencing using a Rapid Barcoding Kit (ONT) and sequenced on an R 9.4.1 flow cell for 24 hours.

Basecalling, alignment, and BrdU calling was performed as described above. A list of differentially methylated CpGs was curated from RRBS methylation profiles for stem and non-stem BT142 cells. CpG methylation status for each nanopore read was calculated using the 'call-methylation' command in nanopolish and reads were assigned to cell state; reads that did not overlap differentially methylated CpGs were discarded as uninformative. Experiment was performed once.

## In vitro cDNA labeling with dU

RNA was isolated from HeLa grown in DMEM/10% FBS via Quick-RNA kit (Zymo) as per manufacturer's protocol, notably including DNase I treatment. First-strand cDNA synthesis was performed using a SuperScript III kit (Invitrogen) as per the protocol (notably including RNase H treatment following cDNA synthesis), except with the supplementation of 1 μM dUTP (NEB) in the dUTP condition. Synthesized cDNA was prepared for sequencing using a Rapid Barcoding Kit (Oxford Nanopore), then sequenced on an R9.4.1 flow cell for 24 h. Basecalling was performed with Guppy.

Basecalled reads were aligned to the human genome (hg38) using minimap2 and reads that aligned to the mitochondrial genome were isolated via samtools. Raw current reads values for each base position

within reads were obtained using the 'eventalign' command from nanopolish. dU scores were obtained in two ways; first through calculation of z-scores at each position, compared to the reference dT and dU current values, and second through calculation of the percentage of a whitelist of fivemers (those with dU current values > 2 standard deviations above the mean dT reference current) that had current values two standard deviations above the reference value. Both measures displayed high concordance. cDNA was sequenced once.

We noticed during the sequencing of dU-containing DNA a common phenomenon of G > A sequencing miscalls, which seemed to occur when the G was surrounded by Ts. At an example position, chrM:11358, the correct five-mer should be TTGTG, which was correctly called in over 90% of control (no dU) reads (Supplementary Fig. 8). However, in the dU-containing reads, this was miscalled as an A in over 40% of the reads (Supplementary Fig. 8). Using our BSPS reference data, we saw that the effect of a dU in positions 2 and 4 of a TTGTG five-mer was a significant elevation of current; indeed these five-mers often had similar current values to a TTATG five-mer. This effect appears to be specific to dU's ability to increase current in positions outside of p3 (see Supp. Fig. 2d); we do not see similar miscalls with BrdU, for instance. Given the major role position 3 plays in base calling, these altered currents induced by adjacent dUs are enough to cause consistent and predictable sequencing errors.

## Mitochondrial proliferation examination

Cytoplasmic hybrid ("cybrid") cells, which are 143B osteosarcoma cells containing either wild type or KSS-associated mitochondrial genomes, were provided by Dr. Brett Kaufman at the University of Pittsburgh. Cells were cultured in DMEM with 10% FBS. For mitochondrial genome labeling, 24 hours following plating, 200 µM BrdU was added to the media. Cells were harvested 24 h after BrdU addition. Cellular DNA was isolated using a Quick-DNA kit (Zymo), prepared for sequencing using a Rapid Barcoding Kit (Oxford Nanopore) and sequenced on an R9.4.1 flow cell for 24 h.

Basecalling was performed by Guppy. Reads were aligned to the human genome (hg38) using minimap2, and reads aligned to the mitochondrial chromosome (chrM) were isolated via samtools. Raw current values were obtained using nanopolish, as described above, and BrdU positivity of reads was determined as described above. Mitochondrial reads were assigned to belonging to either the mutant KSS-associated chrM (chrM$_{MUT}$) or wildtype (chrM$_{WT}$) based on read coverage or deletion of the 5 kb deleted region (chrM:) using the 'mpileup' command in samtools. Reads that did not cover the deletion region were assigned mutational status based on the haplotype of four SNPs that co-segregated with the deletion in reads that covered both (at positions 2991, 3197, 3450, and 6366 in the reference), again by 'mpileup'. Reads that could not be assigned status by either coverage of the deletion or the co-segregating SNPs were discarded as uninformative. Experiment was repeated with three independent cultures of the same cell line, results from each independent replicate are presented as points in Fig. 4f. Significance was calculated by two-sided $t$-test.

## Bacterial co-culture dynamics

T84 immortalized colonic epithelial cells were grown in transwells to allow for cell polarization, and apically infected with *Shigella flexneri* 2a strain 2457 T (or left uninfected as a control). Infected monolayers were incubated in 100 µg/ml gentamycin in DMEM for 24 h to eliminate any residual extracellular *Shigella* and to allow for initial proliferation and cell to cell transfer of bacteria. 24 h post-infection, new media containing 150 µM BrdU (or no BrdU control) was added to the cells, and cells were harvested at the following timepoints: 1 h, 4 h, and 24 h. DNA was isolated from the infected cells using a DNeasy PowerSoil Pro Kit (Qiagen) and prepared for sequencing using a Rapid Barcoding kit. Cells were sequenced on an R9.4.1 flow cell for 24 h.

Reads were aligned to both the human genome (hg38) and the *S. flexneri* 2a strain 2475 T genome (NC_004741.1), and BrdU content was detected as described above. Dam (G$^{6m}$ATC), Dcm (C$^{5m}$CWGG) and $^{5m}$CpG content was called using Guppy (v6.0.7 - gpu) with the "min_modbases-all-context" model and methylation values were extracted using modkit v0.2.4 (both Oxford Nanopore). Reads aligned to the human or *Shigella* genomes were called as being Dam or Dcm methylation-positive with 2 detected methylation events at the specific motifs, and were called as being $^{5m}$CpG-positive with at least 10 detected methylation events at CpG motifs. Reads that did not have at least 2 Dam or Dcm motifs or 10 CpG motifs that could have been methylated were excluded from the analysis. For unaligned reads, values were extracted via 'modkit' and only methylations in the appropriate motif contexts were considered; reads were considered positive with at least one detected Dcm or Dam methylation event.

Genome assembly was conducted using flye v2.9, with the '--nano-hq' parameter. Assembled contigs were visualized using Bandage v0.8.1[41]. Comparison of rotated contigs to reference genome was performed by generating.paf (paired alignment file) using minimap2 and visualized in R. For alignment of T3SS genes, *dcm*, and *dam*, reference gene sequences were downloaded from NCBI and aligned using minimap2. Infection was performed twice per timepoint with independent replicates of the same human cell line and bacterial strain; replicates for BrdU integration analysis are depicted as points in Figs. 5b–d, m. For $^{m6}$A genome assembly (Fig. 5g–l), all infected timepoints from both replicates were pooled and analyzed together.

## Temozolomide characterization

T98G cells (ATCC) were plated in DMEM/10% FBS. 24 h post-plating, 20 µM temozolomide (Selleck) or DMSO was added to the media. 48 h post-plating, 100 µM BrdU was added to the media. 72 h post-plating, cells were harvested, DNA was isolated using a Quick-DNA kit (Zymo) and prepared for sequencing using a Cas9 kit (Oxford Nanopore Technologies) as per the manufacturer's protocol. Briefly, genomic DNA was dephosphorylated and cleaved using Cas9 with two guides on either side of the *MGMT* promoter (Spacer sequences: Upstream_1 5'- AAGCACCTGGCATTCAACCC -3', Upstream_2 5'- GAAGGCAGC TTTGTTGTAAG – 3', Downstream_1 5'- GCCTGCAGAGCTAACAGCGT -3', Downstream_2 5'- AGGCCCTGATTGAGACACCT -3'). Cleaved DNA was then A-tailed and ligated to Nanopore sequencing adaptors, and sequenced for 24 h on an R9.4.1 flow cell.

Basecalling and alignment of sequenced reads was performed as described above. Promoter methylation status for each read aligning to the *MGMT* promoter was calculated using the 'call-methylation' command within nanopolish. For each read, the promoter CpG island methylation was calculated over the following region (hg38): chr10:129,466,800-129,467,600. Methylation upstream of the CpG island (hg38 chr10:129,466,300-129,466,750) was also calculated; reads without methylation of this region were excluded as potentially uninformative newly-synthesized reads with incomplete methylation establishment. Hydroxymethylation status of each read was calculated using Dorado (Oxford Nanopore). BrdU incorporation in each read was calculated as described above. Because the Cas9 sequencing kit is incompatible with barcoding, the DMSO and TMZ treatment conditions had to be sequenced in separate runs and as such were normalized within sequencing runs. Experiment was repeated in triplicate for TMZ treatment and duplicate for DMSO treatment, from the same cell lines cultured at different times and sequenced independently; in Fig. 6e data from individual replicates are depicted as points and significance was calculated via two-sided $t$ test.

In one replicate, during adapter ligation 5 µM of a T-tailed hairpin oligo was added to the adapter ligation mixture. Reads with hairpins were identified via overlapping secondary alignments, and examined for signatures of O6-mG generated by BSPS.

### Cell line availability

The following cell lines from repositories were utilized: LN229 (ATCC CRL-2611), T98G (ATCC CRL-1690), BT142 (ATCC, ACS-1018), T84 (ATCC CCL-248), 293 T (ATCC CRL-3216), 3T3 (Swiss albino) (ATCC CCL-92). The following bacterial strain was utilized in this study: *Shigella flexneri* type 2 A strain 2457 T (ATCC, 700930).

### Data attrition

Only Nanopore sequencing reads which passed quality control filters (read score, length) at the manufacturer's default settings were considered for further analysis. For BSPS synthesis, only reads with the correct number of detectable barcodes and a detectable current at the synthesized *k*-mer were analyzed for current signatures. For multidimensional sequencing analysis, no samples or data points that passed sequencing quality controls (as above) were omitted from analysis.

### Statistics and reproducibility

Specific experimental conditions (sample size, replicate nature, number of replicates) are listed in the appropriate methods section. No statistical methods were used to predetermine sample size. No data were excluded from the analysis, except sequencing reads considered "failed reads" by the manufacturer's default settings, as described above. The investigators were not blinded to allocation during experiments or outcome assessment.

### Box plot characteristics

For box plots (Figs. 2f, 4c; Supplementary Figs. 2d, 2f, and 3h), the center line represents the median value, the box limits are the upper and lower quartiles, the whiskers are 1.5 times the interquartile range, and points represent outlier points outside the interquartile range.

### Reporting summary

Further information on research design is available in the Nature Portfolio Reporting Summary linked to this article.

## Data availability

All processed data are available in the main text or the supplementary materials and data, or the source data file. The raw sequencing reads (Oxford Nanopore fast5 files) generated in this study have been deposited into Sequencing Read Archive (SRA) under accession code PRJNA1083468. An example protocol and example R commands to generate and process a sequenced BSPS library are included as supplementary notes 1 and 2. Source data are provided with this paper.

## Code availability

The code used for recurrent neural network analysis of BrdU is available on GitHub (https://github.com/haibol2016/NanoporeBrdUCaller) and Zenodo (https://doi.org/10.5281/zenodo.15593740)[42]. Example R commands to process and analyze a sequenced BSPS library, or to perform statistical analysis of BSPS data, is included with the supplementary information.

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

## Acknowledgements

We thank J. Dekker, E. Baehrecke, S. Wolfe, M. Kelliher, and A. Mercurio for thoughtful discussion and helpful comments on the manuscript. We thank B. Kaufman (University of Pittsburgh) for the KSS cybrid cell line. We thank L. Robbins and the UMass Chan HPC administrators for computational support. This work was supported by the National Institutes of Health, NCI, NIGMS, and Common Fund (grants K22 CA237846 and DP2 GM159179, both to W.A.F.), and the Sontag Foundation Distinguished Scientist Award (to W.A.F.)

## Author contributions

Conceptualization: C.M.H., B.F.S., B.A.M., L.J.Z., W.A.F. Methodology: S.S.D., B.A.P., K.K., S.V., K.H., H.L., D.L.D., B.F.S., C.M.H., B.A.M., L.J.Z., W.A.F. Investigation: S.S.D., B.A.P., K.K., S.V., K.H., H.L., D.L.D., H.T., B.F.S., L.A. Visualization: H.L., H.T., W.A.F. Supervision: L.J.Z., W.A.F. Writing – original draft: WAF. Writing – review & editing: S.S.D., B.A.P., H.L., C.M.H., B.A.M., L.J.Z., W.A.F.

## Competing interests

The authors declare no competing interests.
