## [Transparent Peer Review file · Nature Communications]

Multidimensional third-generation sequencing of modified DNA bases allows interrogation of complex biological systems.

Corresponding Author: Dr William Flavahan

Version 0:

Reviewer comments:

Reviewer #1

(Remarks to the Author)

In this paper, David et al. introduce a new method for nanopore sequencing of bases beyond the canonical A, C, T, G in a systematic manner. They accomplish this by synthesizing oligonucleotides with all possible sequence combinations of standard bases (A/C/T/G) and a specific modified base (dU, BrdU, O6mG, dl, abasic sites, etc), with a barcode that provides a ground truth for the oligonucleotide sequence. In this manner, when sequencing the oligonucleotides using nanopores, they can identify current patterns that distinguish modified bases from non-modified bases. Since modified and damaged bases are the source of many mutations and also encode epigenetic information, sequencing modified bases holds potential to answer many biological questions. The performance of the method, while not achieving single molecule sensitivity except for abasic sites, is quite good, and the authors demonstrate some very creative proofs of principles for their technology. This includes: a] quantifying cell line sensitivity to chemotherapy by BrdU quantification as a proxy for proliferation rate, b] distinguishing synthesized cDNA from contaminating genomic DNA, c] confirming a known difference in proliferation rate of a specific type of mutant versus non-mutant mitochondria, d] distinguishing a small amount of bacterial DNA from human DNA using differential epigenetic modified bases with much more specificity is possible to distinguish them based on sequence alone, e] simultaneous profiling of DNA sequence, epigenetic methylation state, chemotherapy damage, and cell proliferation rate via BrdU in a glioblastoma cell line.

Overall, this is an excellent and innovative paper highly worthy of publication. Below I provide some important comments, and I think addressing as many of these as feasible will help non-expert readers understand the approach as well as technical readers seeking to expand on this work.

Introduction:

- The authors should also mention enzymatic and antibody-based methods that profile specific modified bases, and disadvantages of these (or advantages) relative to this new paper's method.
- I believe that there are now more RNA modifications that can be detected by Nanopore's commercial/public analysis platform. These should also be mentioned so that this section is up to date. These modifications have become recently possible to analyze due to Oxford Nanopore the company synthesizing training libraries, so if there is some citation for this, that should also be mentioned.

Barcoded Split-Pool Synthesis:

- Why were 5-mers synthesized instead of 6 or 7 mers? Could more long distance secondary structure of bases outside the 5 mer affect the signal even though only ~5-6 mers are inside the pore?
- Would help to describe some estimate of the number of reads/molecules that were profiled in the main text, and what the estimated number is to power the analysis well, in case other groups may want to perform this for other modified bases.
- It would be important to provide in the main text more quantitative metrics on how the final analysis pipeline trained on a subset of the training data discriminates the modified bases on the remaining data: sensitivity, specificity, etc (some metrics giving an estimate of false positive and false negative rate). Although some of these are described in the methods, a short summary would be important to put in the main text. I imagine these metrics may differ depending on the position of the modified base and the number of modified bases in the 5-mer, so the metrics could be specified for one particular situation,

or across all situations.

- The methods indicate that analysis is limited to 5-mers that have above a threshold ROC AUC. I think it would be important to describe in the main text what % of all possible 5-mers are callable for each type of modified base that is analyzed in the paper. Or for more complete characterization, the text could generally describe how changing the ROC AUC threshold affects the % callable 5-mers, which would make the tradeoff between these more explicit.
- Could the modified base affect the efficiency/activity of the type IIs restriction enzyme?
- After a modified base is incorporated in one round of ligation, does the overhang of the donor oligo in the next round of ligation need to hybridize to the prior modified base? If I understand this correctly, could this bias to some degree the obtained oligos due to inefficiencies in hybridization and ligation to the modified base?
- In Fig. 1F: what is the reason for the relatively low yield (11.8%) of 5bc,5mer molecules?
- If I understand correctly, the authors call a specific modification based on its current difference relative to a specific non-modified base that corresponds to the modification. For example, BrdU current is compared to dT current. Does the analysis assume that Nanopore's base calling of A/C/T/G always labels brdU as dT? Ext Data Fig 7 shows that nanopore base calling software can miscall dU as something other than dT for example -- will the current analysis approach miss modified bases that are miscalled? What are the data/metrics regarding miscalling for other modified bases other than dU?

Tracking in vivo and in vitro nucleotide synthesis using two-dimensional sequencing

- It might help to add a sentence to describe the motivation of performing an experiment co-culturing more than one cell line. Usually in drug sensitivity screens, different cell lines would be screened separately. Was the reason for doing this just for this validation experiment, or is this rather showing a proof of principle for potentially future experiments that mix together many cell lines in competition experiments for example?

Multidimensional sequencing permits single-molecule interrogation of chemotherapy response

- I suggest rephrasing the following in terms of selection forces rather than ascribing a motive to the cells. "We hypothesized that some MGMTme cells would attempt to remove the inactivating methylation in order to re-express MGMT and repair the damage"
- The rationale for analysis of 5hm-C as an indication that MGMT promoter methylation was not reversed should be clarified for non-expert readers.
- This section describes a hairpin. It would help to clarify earlier here or earlier in the paper what this hairpin refers to.
- I'm surprised that only a single read containing O6mG was identified. Is this within the range of expectation from known pharmacodynamics and dosing? Were only reads from MGMT promoter enriched libraries analyzed for this?

Discussion:

- Could a future analysis achieve greater performance by integrating information as a modified base proceeds across the nanopore from the entrance to the exit, rather than only using data from one position in the nanopore? (I assume that is not currently possible since the barcode encodes the identity of the bases only for the synthesized 5-mer.)

Figures:

- Ext Data Fig 1: legend typo: "Schematic depicts even an odd barcode strategy."
- Ext Data Fig 1a: the overhang is 4 bp but schematized as 2 bp.
- Ext Data Fig 1a: the use of odd and even is a bit confusing since it implies different length overhangs (odd and even), but I think the authors mean odd and even ligation cycles. That could be clarified better in the text.

Reviewer #2

(Remarks to the Author)

David et al., create a clever, powerful approach to generating training data sets for Oxford Nanopore detection of modified bases. This appears both cost effective and overcomes limitations with other approaches, especially when it's challenging to generate specific modifications with high frequency. They generate reference oligos and current data of multiple modified bases and then use these data to show detection of modified bases in multiple biological contexts including DNA replication, in vitro synthesis, mtDNA replication, human pathogen infection, and in the dynamics of a chemotherapy response in glioblastoma cells. The utility of the approach. This is a powerful resource for the field. However, I think additional steps could be taken to making analysis of new Nanopore data more accessible such as providing scripts for their modified basecalling and Monte Carlo simulations. In addition, it would be informative to see how their approach benchmarks compared to some of the latest models present in Dorado and the latest DNAscent versions. Overall, this is a well-executed strategy and a well-written paper with clear figures.

Major comments

- The technology discussed is both a powerful way to train new models in a cost-efficient way. The data generated are also an incredibly useful resource for the community. The protocol for the barcoded split-pool synthesis seems well explained. However, additional steps could be taken for making the analysis of data more accessible to users. The parameters of basecalling and alignment seem clear, but the details of calling modified bases less so. The methods state that no custom code or computer programs were created. Providing some custom code for basecalling and for the Monte Carlo simulations seems worth it to make this an even more accessible resource to the field, even if only as example code that could then be modified by the user.
- The authors discuss their approach as overcoming the limitations that other approaches have had, such as for bases that cannot be generated with high frequency. They go on to say that nanopore detection is limited to certain modifications and there are limitations. While I agree with all of this and agree their approach clearly has potential to overcome these

limitations, the authors don't show benchmarks of their approach compared to these other methods. It would be beneficial to show comparison for the modifications that have existing pipelines such as 5-methylcytosine, 6-methyladenosine, and BrdU.

For example, Extended Data Figure E5 shows DNAscent's analysis of BSPS BrdU oligos. How does this compare to the calling of BrdU from the current profiles generated by BSPS oligos? In addition, the authors are using DNAscent version 3.1.2. Are there significant differences with the latest versions of DNAscent (v4.0.3)?

In another example, how does the 6-methyladenosine calling compare to that of the latest Dorado high accuracy basecalling models? As part of that question, is there a reason the authors are using Guppy rather than Dorado?

Minor comments

- Lines 14-15: "a myriad of chemical modifications" should be "myriad chemical modifications"
- Figure 1F: It's not clear to me if the three shades of green/gold are denoting something different. At first I thought they aren't but they aren't the same widths. This should be clarified or simplified to just one shade each.
- Figure 2D-2E and Lines 122-131 – Based on the result, I'm assuming the non-stem cells are slower/less proliferative, but it might be worth briefly mentioning the biology of the system in the text when it's introduced
- Lines 134-145 – should specify "human" mitochondrial genome for clarity
- Figure 2J: The text of "forward" and "reverse" is too small
- In several places throughout the text, the authors use the language of "fraction of genomes divided". Is there a reason the authors are using this language instead of "replicated"? (e.g. Figures 3F and 4C)
- Figure 4G: What are the 13% of reads that have no m6A but don't map to Hg38? I believe the text on line 223 indicates that these are likely repetitive sequences that may not align well in Hg38? Not strictly necessary for your point, but have you tried aligning to the T2T assembly instead?
- Lines 132-149: It's really cool that this can be a way to track synthesized DNA in vitro, and it's a nice validation of mtDNA transcription the way the cDNA aligns. For most practical purposes, I think people are going to use DNase to remove contaminating DNA from samples before doing cDNA synthesis. It might be worth discussing here or in the discussion situations where this strategy could be used.
- The text on lines 192-194 discusses that the infected conditions have less replication than the uninfected controls and refers to Figure 4C. The figure doesn't seem to distinguish between infect and uninfected human samples.
- Line 247: I think "the gene encoding this gene" should be "the gene encoding this protein"

Reviewer #3

(Remarks to the Author)
Notes on David et al. 2025

Comments for the author

In this manuscript the authors introduce a novel method to produce all possible unique kmer oligos of a specific length (barcoded split-pool synthesis; BSPS) with a base modification. This is essential to support detection of modification using third generation sequencing technologies, such as ONT and Pacbio sequencing. The approach is used to determine signal levels of a specific modification in all sequence contexts and subsequently used in a number of use cases, such as screening drug sensitivity or studying mitochondrial genome dynamics. The work represents a very useful demonstration of split-pool synthesis to the detection of DNA modifications using 3rd gen sequencing platforms, which is an important advance of the field that undoubtedly will find great applications. However, given its presentation as a method there is limited attention to the method development and validation, in lieu of practical demonstration.

Major:

- The authors extensively use Nanopolish. Nanopolish eventalign gives "the optimal alignment" between the measured raw current signal and the expected segment values, where the latter is taken from a lookup table that has a value assigned to all possible k-mers. Aligning the raw signal (with a modification) to this expected sequence (without any corrections done for modifications) may lead to the aligner producing a result where the values for the modified base are assigned to the neighbouring base (/overlapping k-mer) instead. Hence, looking for differences in value distributions can be difficult as the actual signal change is now mis-aligned. Nanopolish itself gets around this issue when calling methylation by employing an HMM in the region around the target motifs, hence effectively allowing the HMM to find a better local realignment solution if needed, where the alignment with both a modified and unmodified alternative of the same sequence are tested and the most likely option to have produced the measured current is called. The approach employed by the authors seems more to the approach employed in Tombo (which by the way is also not optimal).
- In relation to the point above: why did the authors not explore one of many deep learning solutions? They themselves call their data 'training libraries', so why not leverage RNN-based models which get around the issue of relying on alignment

only?

- For the R9 pore model it is generally assumed that the signal is influenced by 5 or 6 bases in the pore at any one time, hence 5-mers and 6-mers seem a fitting k-mer length. However, if a 5-mer influences the signal, calling a modification from just the 5-mer where the target base is in the very center ignores information that can be obtained from up to two bases to the left and right. This becomes especially important for pores that have a wider read head, such as R10. Have the authors considered this in their analysis, and might this improve the results?

- I think the estimation of the performance of the approach should be extended and not hidden in an external figure.

Moreover, the 'per read' performances obscure the 'per base' performances (which are in E5). I would expect from a new method that there is an estimated error rate (at a per-base rather than per read level) of the method and that there is a comparison to the sota method on a gold standard or positive control. The suggestion of high accuracies presented in E4K and L (based on only a few 'callable kmers' with modest accuracy) seems to suggest that most signal is due to the assumption several 'callable' kmers are present in a read and not due to high accuracy of the method itself. In sum, the "statistical analysis" on the bottom of p23, deserves more attention and clarity. Of note, ROC analysis in general is not very suitable for situations where the vast majority of calls is expected to be negative, as just calling everything negative will already yield a good performance. Precision/recall/FDR estimates becomes especially relevant in case of rare modification, as in that case a non-negligible error rate can already give rise to very high false discovery rates (FDRs). It is unclear if the presented performance also holds true for the other proposed base modifications / analogs. If they can show that this BSPS provides a higher (modified) basecalling accuracy / lower FDR this would really endorse their point of the utility of this method.

- The authors rightfully point out that this approach could be the basis for basecalling methods for other pore designs, presumably hinting at the more commonly used R10 pore, which has a much longer read head. This will require many more split pool 'cycles'. It would be good to be more concrete about the feasibility of such approach and how much coverage would be expected when kmers of e.g. 40 bp are required for model training.

Minor

Pg21 l45: "the manufacturer's protocol starting from the 'Adapter ligation and clean-up' step" needs a version or something else much more specific (perhaps written elsewhere?)

Pg22 l7: "Reads that fully mapped", is that reads that just map from start to end, without any hard or softclipping (rare)? Or just reads that did not get split mapped?

Pg22 l12-l21: I don't see how this could be done without any code.

Pg22 l20-l21: This noise correction is not explained

Reviewer #4

(Remarks to the Author)

David et al. introduced Barcoded Split-Pool Synthesis (BSPS), which is a rapid and inexpensive experimental technique to produce reference standard, modification-containing DNA oligos. The BSPS technique is uniquely suitable for producing training data for nanopore sequencing-based modification detection algorithms. Leveraging BSPS, the authors studied various modifications, e.g. BrdU, under a broad spectrum of biological processes, e.g. DNA replication.

Producing ground-truth training data is of crucial importance for nanopore sequencing-based modification detection, and is the latest focus of the nanopore community. I therefore consider this paper to be highly significant. My comments are:

1. Preventing self-ligation is essential, and the "even-odd" design is smart. Therefore, I would suggest the authors to add a schematic panel in Figure 1A to highlight the design.
2. If I understand correctly, added bases correspond to barcodes. However, in Data S1, only 7 barcodes were provided, without noting their corresponding bases. Would the authors please clarify?
3. Also for Data S1, if I understand correctly, "r1" are round 1 donor reverse sequences. "N" in "n" donor reverse sequences will basepair with the modification added in the previous round. Would the authors please clarify? Meanwhile, "terminators" were not cited in the manuscript: will "V3-5term_hp_odd" and "V3-5term_hp_even" be added to the barcode side, while the "V3-3term_hairpin" to the base side when BSPS finishes? Are they also added through ligation? If so, why there are no "F" and "R" sequences?
4. Considering the above complications, I would suggest the authors provide the .fa file for "all possible fully-synthesized oligos", or a script for enumerating such combinations, to make the BSPS bioinformatics user-friendly.
5. I am wondering how efficient the BSPS ligation is. Maybe the authors can quantify the oligo length distribution using nanopore sequencing reads, as the quality control for BSPS?
6. Typically, R9.4.1 DNA nanopore sequencing signal events are corresponded to 6mers. For example, nanopolish (an essential bioinformatic component in this study) uses 6mer model to perform "eventalign". I totally acknowledge that BSPS can produce 6mer training data, but could the authors please clarify the rationale behind producing 5mers?

7. Related to the previous comment, from a technical perspective, could the authors please describe how the 5mer event tables were produced, as nanopore's eventalign produces 6mer event tables by default?

8. DNA regions containing (particularly dense, such as BSPS oligos with 5 consecutive BrdUs) modifications might completely disrupt basecalling therefore producing "unmappable reads". As a result, "characteristic modification signals" will not be recorded in nanopore's eventalign event tables: only signals fall within "canonical signal distributions" will be recorded. Did the authors observe decreased aligned read% for densely modified oligos?

9. The R9.4.1 chemistry has been discontinued and the R10.4.1 chemistry is the state-of-the-art. Therefore, discussing the compatibility between BSPS and R10.4.1 may further expand the impact of this paper.

10. RNA modification detection emerges as the state-of-the-art research focus of the nanopore sequencing community. Therefore, discussing the possibility of producing RNA oligos may further expand the impact of BSPS.

11. No sure what does "two/multi-dimensional sequencing" mean. Maybe the authors referred to multi-omics (sequence + modification)? Maybe consider using "native DNA sequencing"?

12. For the METHODS section "Statistical analysis of modified base detection", it is important to clarify the number of training samples generated for each of the 2101 5mers. Specifically, it would be useful to know whether the training samples are evenly distributed across all the 5mers or if there is any data imbalance.

Reviewer #5

(Remarks to the Author)

Version 1:

Reviewer comments:

Reviewer #1

(Remarks to the Author)

The authors have satisfactorily addressed all of my comments. I expect this paper will have a strong impact and will spur many others in this research field.

(Remarks on code availability)

Reviewer #2

(Remarks to the Author)

In the revised version of the manuscript, the authors have addressed my major concerns. The addition of a detailed supplementary protocol (and numerous scripts) for both BSPS library synthesis and data analysis, which should make this approach much more accessible to potential users. The authors also trained an RNN model on their BrdU data and incorporated this in their comparison of BrdU detection with DNAscent. Outside of these significant changes, there were numerous smaller tweaks that improve clarity throughout, as well as the addition of an additional extended figure detailing potential expansions of the approach. I had one minor comment, which is that Extended Data Figure E6A, the image quality of the added matrices is too low and they aren't legible (it looks good in the Python notebook).

(Remarks on code availability)

Scripts are organized and divided between a Python notebook overviewing RNN training and scripts for processing ONT sequencing data. I was not able to test all of the scripts, but they all seem relatively straightforward and should be easy to pick up for someone with some basic bioinformatic and ONT pipeline familiarity.

Reviewer #4

(Remarks to the Author)

The authors have addressed all my questions.

(Remarks on code availability)

NA

Reviewer #5

(Remarks to the Author)

(Remarks on code availability)

Response to Reviewer comments:

Summary:

We want to thank the reviewers for their thoughtful comments on our manuscript submission, “Multidimensional third-generation sequencing of modified DNA bases allows interrogation of complex biological systems,” by David, Pacheco et al. We are very grateful that the reviewers each recognized the innovation and potential utility of our method, and thank the reviewers for their thoughtful suggestions, which have made the revised manuscript significantly stronger than the initial submission.

Several general comments prior to the specific responses to each reviewer: multiple reviewers asked about more advanced computational analysis of our data, including through machine learning approaches. In this revision, we now include results from a recurrent neural network (RNN) model trained on one of our barcoded split-pool synthesis (BSPS) libraries, which shows enhanced base detection at the single base level and highly similar concurrence at read-level detection with existing methods (see new Extended Data Figure E6). Additionally, there were several comments requesting clarification about specific steps in our synthesis approach and its analysis; in response we have included an entire protocol for the synthesis and analysis of a BSPS library, which should greatly assist a reader who wishes to perform this method for a novel base. There were also questions about discussing the potential adaptability of this method to the newest generation of nanopore flow cells (the R10); this is an area of active investigation in our lab and we have now included a new Extended Data Figure discussing potential modifications for the R10 and future cells.

We are very grateful to the reviewers for their thoughtful comments and suggestions; we feel that this version of the manuscript has greatly enhanced the accessibility of our work and the potential for readers across the biological sciences who might be interested in performing modified base sequencing to enhance their work with our methods and available data.

Specific responses to reviewer comments are as follows:

Response to Reviewer #1:

In this paper, David et al. introduce a new method for nanopore sequencing of bases beyond the canonical A, C, T, G in a systematic manner. They accomplish this by synthesizing oligonucleotides with all possible sequence combinations of standard bases (A/C/T/G) and a specific modified base (dU, BrdU, O6mG, dl, abasic sites, etc), with a barcode that provides a ground truth for the oligonucleotide sequence. In this manner, when sequencing the oligonucleotides using nanopores, they can identify current patterns that distinguish modified bases from non-modified bases. Since modified and damaged bases are the source of many mutations and also encode epigenetic information, sequencing modified bases holds potential to answer many biological questions. The performance of the method, while not achieving single molecule sensitivity except for abasic sites, is quite good, and the authors demonstrate some very creative proofs of principles for their technology. This includes: a) quantifying cell line

sensitivity to chemotherapy by BrdU quantification as a proxy for proliferation rate, b] distinguishing synthesized cDNA from contaminating genomic DNA, c] confirming a known difference in proliferation rate of a specific type of mutant versus non-mutant mitochondria, d] distinguishing a small amount of bacterial DNA from human DNA using differential epigenetic modified bases with much more specificity is possible to distinguish them based on sequence alone, e] simultaneous profiling of DNA sequence, epigenetic methylation state, chemotherapy damage, and cell proliferation rate via BrdU in a glioblastoma cell line.

Overall, this is an excellent and innovative paper highly worthy of publication. Below I provide some important comments, and I think addressing as many of these as feasible will help non-expert readers understand the approach as well as technical readers seeking to expand on this work.

We thank the reviewer for the recognition of our work, and are extremely grateful for the advice directed towards helping non-expert readers understand. Our main goal with this manuscript is to create a tool that's useful for people who aren't necessarily already doing third-generation sequencing, which is why we've submitted to *Nature Communications* instead of a more technically-focused journal.

Introduction:

- The authors should also mention enzymatic and antibody-based methods that profile specific modified bases, and disadvantages of these (or advantages) relative to this new paper's method.
- I believe that there are now more RNA modifications that can be detected by Nanopore's commercial/public analysis platform. These should also be mentioned so that this section is up to date. These modifications have become recently possible to analyze due to Oxford Nanopore the company synthesizing training libraries, so if there is some citation for this, that should also be mentioned.

We have incorporated these suggestions – while the original text focused on DNA, adding the RNA makes sense and is a great way to highlight that the potential for detecting DNA modifications exists, similar to how the RNA cells can call RNA modifications.

Barcoded Split-Pool Synthesis:

- Why were 5-mers synthesized instead of 6 or 7 mers? Could more long distance secondary structure of bases outside the 5 mer affect the signal even though only ~5-6 mers are inside the pore?

This is an excellent question. The choice of 5-mers versus 6+ is based on the tradeoff of coverage versus precision. With increasing k-mer length, the possibility space increases (e.g. there are 3,125 (5^5) 5 base 5-mers, 15,625 (5^6) 5 base 6-mers, and 78,125 (5^7) 5 base 7-mers), and our initial analysis of a 6-mer library we had constructed was that the slight increase in accuracy was not worth the decreased per-mer coverage, in terms of cost per sequencing depth. One main advantage of our BSPS protocol, however, is that it can be run for as many rounds as desired – a scientist reading this paper with the desire and funds could easily synthesize, sequence, and analyze 7-mers containing any modified base for their own work. Additionally, the protocol can be modified to add multiple bases at a time to more rapidly and efficiently synthesize longer k-mers with the trade-off of an increased upfront cost in donor and oligo barcode generation – we've added an additional extended data figure (new **Ext Data Fig E10**)

to demonstrate potential adaptations to the method to easily generate longer k-mers, and to introduce randomness outside of the synthesized k-mer to increase the robustness of the method.

- Would help to describe some estimate of the number of reads/molecules that were profiled in the main text, and what the estimated number is to power the analysis well, in case other groups may want to perform this for other modified bases.

This is a great point – while some of this information was included in the extended data figures for each modified base (e.g. Fig 1F, Ext Data Fig 2B), we have compiled this information together into a single figure with a specific callout in the main text, now presented as **Extended Data Table E2**.

- It would be important to provide in the main text more quantitative metrics on how the final analysis pipeline trained on a subset of the training data discriminates the modified bases on the remaining data: sensitivity, specificity, etc (some metrics giving an estimate of false positive and false negative rate). Although some of these are described in the methods, a short summary would be important to put in the main text. I imagine these metrics may differ depending on the position of the modified base and the number of modified bases in the 5-mer, so the metrics could be specified for one particular situation, or across all situations.

- The methods indicate that analysis is limited to 5-mers that have above a threshold ROC AUC. I think it would be important to describe in the main text what % of all possible 5-mers are callable for each type of modified base that is analyzed in the paper. Or for more complete characterization, the text could generally describe how changing the ROC AUC threshold affects the % callable 5-mers, which would make the tradeoff between these more explicit.

In response to these and comments from other reviewers, we have added a recurrent neural network (RNN) trained on the BrdU data from our BPS synthesis, and have added a section to the text regarding the modified base distinguishing ability. Using 80% of the BPS oligos for RNN training, we achieved a 83.9% sensitivity and 78.8% single-base specificity on the remaining 20% of BPS oligos (data now presented in a new **Extended Data Figure E6**).

One other point that I want to highlight (and have attempted to make clear in the text) is that the percentage of callable 5-mers is not necessarily a limitation of the BPS approach – in many cases, it is actually an estimation of the inherent limitation of the nanopore’s ability to distinguish these bases, as estimated by our BPS data. It is possible that deeper sequencing of a BPS library or a more sophisticated computational approach could slightly increase the fraction of callable 5-mers (or the single-base detection ability in a way not reliant on subsetting to callable 5-mers), but at some point the reason a 5-mer would be “not callable” is fundamentally because the difference in chemical structure of the base modification in the context of that specific 5-mer does not cause a difference in the current flowing around that molecular structure within the nanopore during sequencing.

- Could the modified base affect the efficiency/activity of the type IIs restriction enzyme?

- After a modified base is incorporated in one round of ligation, does the overhang of the donor oligo in the next round of ligation need to hybridize to the prior modified base? If I understand this correctly,

could this bias to some degree the obtained oligos due to inefficiencies in hybridization and ligation to the modified base?

These are excellent questions – one of the ways we test this is presented in Extended Data Figure E1D, where we look at the read coverage of 5-mers with increasing amounts of the modified base, and for any of the bases included in this manuscript, we don't see significant undercoverage as representation increases, which would suggest a systemic problem such as inability to ligate or hybridize. However we do think this may present issues with some other bases - we had originally attempted to also generate a modified base library for 8-oxo-dG, but our analysis suggests that there was a problem with, as the reviewer suggests, either ligation to or cleavage of this base from the developing oligos. While this is problematic for the synthesis protocol as presented, we now present this problem and a potential simple workaround for problematic bases (such as 8-oxo-dG) explicitly in new **Extended Data figure E10** (panel C). By using donor oligos that add multiple bases, so that the modified base is not the adjacent base for restriction or ligation, we believe this problem can be bypassed, with a slight increase in complexity of the library synthesis protocol.

- In Fig. 1F: what is the reason for the relatively low yield (11.8%) of 5bc,5mer molecules?

We believe this is due to the cumulative incomplete ligation efficiency. With the addition of 5 barcodes, 5 donors, and two endcap hairpin terminators, even a high but not complete ligation efficiency will result in final yield rates around what we see ($85\%^{12} = 14\%$, for instance). Even minor further gains in ligation efficiency (if, for instance, NEB releases a further-optimized blunt/TA ligation kit) could dramatically increase the final success rate.

- If I understand correctly, the authors call a specific modification based on its current difference relative to a specific non-modified base that corresponds to the modification. For example, BrdU current is compared to dT current. Does the analysis assume that Nanopore's base calling of A/C/T/G always labels brdU as dT? Ext Data Fig 7 shows that nanopore base calling software can miscall dU as something other than dT for example -- will the current analysis approach miss modified bases that are miscalled? What are the data/metrics regarding miscalling for other modified bases other than dU?

One quick minor note – in the figure that was Ext Data Fig E7 in the original submission (and is now Ext Data Fig E8), the error is not that the dU is miscalled as a non-dT, it's that an adjacent dG is more frequently miscalled as a dA – the dU is called as a dT as would be expected. This seems to be a common problem with the R9, which was that many dA and dG k-mers had similar reference currents, so the change induced by a dU at position 4 was enough to confuse the caller about which purine was present at position 3. (The introduction of the second pore in the R10 appears to help mitigate this problem, since dA/dG have strong distinguishing effects when in positions 1-4, in the first pore of the new two-pore setup).

This is a great question though - our analysis pipeline is based on reference-corrected sequencing data so is relatively resistant to this problem (as the observed current values are close enough to align to either a dA or a dG in the reference), unless the current difference induced by the modified base is very large. For instance, every single abasic oligo is incorrectly basecalled around the abasic site, and is just called as

“NNNNN” by nanopolish. Anecdotally, BrdU and dU are almost always called as dT (although dU does sometimes affect the calling of nearby purines, as shown), both 5-methyl- and 5-hydroxymethyl-dC are almost universally correctly called as dC, and 6-O-methyl-dG is very often called as dA, which can be corrected for in our pipeline quite simply. Unfortunately, I don’t think we can derive exact quantifications of these rates, since filtering of the oligos as part of our BSPS analysis is stringent to exclude incorrectly synthesized 5-mers, and our experimental data contains stochastically incorporated bases so we don’t have a ground truth to compare the BSPS basecalls to. As our analysis of the abasic sites shows, though, this problem can be corrected computationally through careful selection of the reference bases and careful analysis of the results and the success rate of alignment.

Tracking in vivo and in vitro nucleotide synthesis using two-dimensional sequencing

- *It might help to add a sentence to describe the motivation of performing an experiment co-culturing more than one cell line. Usually in drug sensitivity screens, different cell lines would be screened separately. Was the reason for doing this just for this validation experiment, or is this rather showing a proof of principle for potentially future experiments that mix together many cell lines in competition experiments for example?*

This is a great suggestion – we’ve added a line to the text to put these results in better context. Realistically, as the reviewer suggests, this specific experiment in this figure is more of a proof-of-concept showing separation of reads that contained modified base information by genetic sequence. The specific case of two cell lines is not likely to be widely useful (as mentioned, these can easily just be screened separately), but a similar experiment involving either cell line cooperation or competition (as suggested), or something like finding genetic differences due to CRISPR edits or a subclonal oncogene mutation, could function on the same principle. Our main rationale in designing the specific experiment was the availability of ground truth – we knew one cell line was resistant and one was sensitive so we could compare our results to the known data to show the approach was valid, which suggest that similar approaches based on the same underlying principle will also likely be successful.

Multidimensional sequencing permits single-molecule interrogation of chemotherapy response

- *I suggest rephrasing the following in terms of selection forces rather than ascribing a motive to the cells. “We hypothesized that some MGMTme cells would attempt to remove the inactivating methylation in order to re-express MGMT and repair the damage”*

- *The rationale for analysis of 5hm-C as an indication that MGMT promoter methylation was not reversed should be clarified for non-expert readers.*

These are great points and we have made these clarifications to the text.

- *This section describes a hairpin. It would help to clarify earlier here or earlier in the paper what this hairpin refers to.*

This is a great point – while the hairpin was described in the methods, we have clarified in the main text how the hairpin added to the end of the read affects sequencing.

- I'm surprised that only a single read containing O6mG was identified. Is this within the range of expectation from known pharmacodynamics and dosing? Were only reads from MGMT promoter enriched libraries analyzed for this?

Based on previously published data (Stratenwerth et al., *Mol Cancer Ther.* 2021, PMID 34253592) showing an O6mG frequency of about 1-2 in 10^6 bases at this dose of TMZ, and our sequence coverage of about 3 million bases at the MGMT promoter, we would expect to see about three to six(-ish) O6mGs. We did identify one or two other bases that had current values that could have been consistent with O6mG, but the current difference caused by O6mG at those two 5mers is less dramatic, and none of those reads had a successfully-ligated hairpin to confirm the opposite-strand sequence, so we were not confident enough in those data to present them in the manuscript. We did only analyze the MGMT promoter-enriched libraries, and within those libraries we only analyzed the reads that specifically mapped to the promoter of MGMT. Notably, the Cas9 sequencing kit reduces the overall yield of a nanopore sequencing run, and while the per-base enrichment of the targeted area over a standard sequencing kit is dramatic (~2,500-fold in our experiments), the absolute number of on-target reads is still low, on the order of a couple hundred per sequencing run. Our reasoning for only analyzing promoter reads here was that the high coverage of these areas allowed us to rule out a SNP or even just a weird-sequencing area/nucleotide sequence – we had several hundred other readings of the exact position where we think the O6mG is present in the read shown in figure 5 (and even more in the non-TMZ treated control libraries) to confirm that this specific current reading is indeed an outlier specific to this read. An approach specifically designed to find O6mG residues, using a different library preparation strategy and ability to control for site-specific sequencing signals, would likely find more.

Discussion:

- Could a future analysis achieve greater performance by integrating information as a modified base proceeds across the nanopore from the entrance to the exit, rather than only using data from one position in the nanopore? (I assume that is not currently possible since the barcode encodes the identity of the bases only for the synthesized 5-mer.)

We agree with the reviewer that this is likely possible. While each synthesized oligo only contains one k-mer, we also have data for the new k-mer that would be formed by base advancement in a separate read – these could be combined to track a modified base as it transits the pore in each position of the fivemer. The approach we've largely utilized in the paper is a relatively simple z-score calculation to determine which reference distribution an observed value most likely belongs to. For BrdU, the largest difference between the modified and unmodified base almost always occurs when that base is in position 3, and BrdU substitutions in other positions do have effects (see Ext Data Fig E2E), but the magnitude of those effects is smaller than the noise of the sequencer so adding them doesn't seem to increase the detection capability of our simple test. For other bases this is not necessarily true – dU often causes a very strong effect on current in position 4 (see Ext Data Figures E2C and E7E), and even a relatively simple analysis could make use of multiple positional information. Additionally, the new RNN-based approach displays higher single-base accuracy on the withheld BPS data used as a validation set, confirming that more complex approaches are likely more accurate at the single-base level. This increased single-base

accuracy does not seem to be required for the read-level analyses performed in the experiments presented in this manuscript however; the RNN and the simple analysis were in agreement on nearly all classified reads in real data (~99.5% agreement, see new **Ext. Data Fig E6**).

Figures:

- *Ext Data Fig 1: legend typo: "Schematic depicts even an odd barcode strategy."*
- *Ext Dat Fig 1a: the use of odd and even is a bit confusing since it implies different length overhangs (odd and even), but I think the authors mean odd and even ligation cycles. That could be clarified better in the text.*

Thank you for the clarity and typo suggestions, we have made these corrections.

Response to Reviewer #2:

David et al., create a clever, powerful approach to generating training data sets for Oxford Nanopore detection of modified bases. This appears both cost effective and overcomes limitations with other approaches, especially when it's challenging to generate specific modifications with high frequency. They generate reference oligos and current data of multiple modified bases and then use these data to show detection of modified bases in multiple biological contexts including DNA replication, in vitro synthesis, mtDNA replication, human pathogen infection, and in the dynamics of a chemotherapy response in glioblastoma cells. The utility of the approach. This is a powerful resource for the field. However, I think additional steps could be taken to making analysis of new Nanopore data more accessible such as providing scripts for their modified basecalling and Monte Carlo simulations. In addition, it would be informative to see how their approach benchmarks compared to some of the latest models present in Dorado and the latest DNAscent versions. Overall, this is a well-executed strategy and a well-written paper with clear figures.

We thank the reviewer for these comments and the appreciation of our manuscript, and especially for the constructive suggestions to improve our manuscript.

Major comments

- *The technology discussed is both a powerful way to train new models in a cost-efficient way. The data generated are also an incredibly useful resource for the community. The protocol for the barcoded split-pool synthesis seems well explained. However, additional steps could be taken for making the analysis of data more accessible to users. The parameters of basecalling and alignment seem clear, but the details of calling modified bases less so. The methods state that no custom code or computer programs were created. Providing some custom code for basecalling and for the Monte Carlo simulations seems worth it to make this an even more accessible resource to the field, even if only as example code that could then be modified by the user.*

This is a very fair comment. In order to increase the ability for users to fully utilize our method, we have now included both the actual benchwork protocol and the example R commands (including the Monte

Carlo commands) as a supplementary data file, in order to assist users with the generation and analysis of BSPS data. One thing to highlight is that thanks to the easy-to-use community tools such as nanopolish, the processed data are actually quite simple to process – the R commands are rather basic.

• *The authors discuss their approach as overcoming the limitations that other approaches have had, such as for bases that cannot be generated with high frequency. They go on to say that nanopore detection is limited to certain modifications and there are limitations. While I agree with all of this and agree their approach clearly has potential to overcome these limitations, the authors don't show benchmarks of their approach compared to these other methods. It would be beneficial to show comparison for the modifications that have existing pipelines such as 5-methylcytosine, 6-methyladenosine, and BrdU. For example, Extended Data Figure E5 shows DNAscent's analysis of BSPS BrdU oligos. How does this compare to the calling of BrdU from the current profiles generated by BSPS oligos? In addition, the authors are using DNAscent version 3.1.2. Are there significant differences with the latest versions of DNAscent (v4.0.3)?*

With the revision of the manuscript, we have included a recursive neural network (RNN), which when trained on a training subset (80%) of BSPS oligos is able to call BrdU on the remaining validation subset (20%) with a sensitivity of 84% and a specificity of 79%. We have added this method to the direct comparison between DNAscent and our basecalling detection is provided on sample data in extended data figure E4K and E4L and **new Extended Data Figure E6**.

Based on release notes (<https://dnascent.readthedocs.io/en/latest/releaseNotes.html>), our understanding is the difference between DNAscent versions 3.1.1 and 4.0.3 is only the inclusion of R10 support in v4+, and that calling of data generated on an R9 should not be different between the models. We attempted to analyze our data with both DNAscent versions 4.0.1 and 4.0.3, but kept getting input/output errors, stating that the HDF5 “internal path traversal failed.” We were unable to resolve this issue using ONT's fast5 and pod5 APIs to convert the file to different formats. I suspect that the internal file structure of an R9/R10 raw file is different, or other options must be specified to analyze R9 data in version 4+ (that I didn't find in the documentation). Fundamentally though, I think the primary limitation of the accuracy is the actual single-base discrimination of the pore. For most contexts, there is simply significant overlap between the currents of the BrdU and the dT, and for some fraction of the time the current values from a BrdU will be simply indistinguishable from the current values expected from a dT. I think the most dramatic increase to modified base detection accuracy will likely be continued iteration from ONT on the engineering of the pore characteristics, rather than more sophisticated analysis techniques; I think one of the advantages of our oligonucleotide synthesis method is that the BSPS libraries provide a way to assess what these pore limitations are. With the revision, we have now added a panel to Ext Data Figure E5 (panel C) to show the overlap in current profiles from each of these 5-mers, which should serve to highlight that the likely limit to the theoretical discrimination potential of any analysis method, BSPS-based or otherwise, is likely with the measurements of the pore rather than the computational analysis of those measurements.

In another example, how does the 6-methyladenosine calling compare to that of the latest Dorado high accuracy basecalling models? As part of that question, is there a reason the authors are using Guppy rather than Dorado?

We apologize for the lack of clarity in the manuscript, but we did not generate a 6mA BSPS dataset – given that we are only examining 6mA in the dam motif (GATC), the main advantage of BSPS is removed. With a fixed four-base motif, there is not a large k-mer possibility space like there is for e.g. BrdU, which could be present at a dT in any 5-mer context. The 6mA calling presented in figure 4 was performed with guppy (in contrast to the BrdU, which was called with our data as in the other experiments); the 6mA data here is presented to demonstrate the utility of base modification sequencing rather than a specific additional advantage of BSPS.

As to the specific usage of guppy over dorado, we were unable to find a 6mA model for the R9 flow cell in dorado (the dorado supported models are listed here, and only include 6mA for R10 DNA flow cells: <https://github.com/nanoporetech/dorado?tab=readme-ov-file#dna-models>). For this analysis, we used the “res_dna_r941_min_modbases-all-context” guppy model, which includes 6mA calling. For non-6mA analyses, nanopolish aligns currents to the reference rather than the basecalls, and given the tolerance of minimap2 for sequencing errors, we don’t think switching from guppy to dorado (most of the advances of which focus on the R10 flow cells) would recover significantly more reads of successfully synthesized oligos or significantly improve the results of our analysis. An additional reason for the usage of guppy is that, at present, SRA will only accept raw nanopore sequencing data in the “basecalled fast5 format”, which is no longer actually supported by Oxford Nanopore and it is not possible to generate this format using dorado, requiring us to use guppy as the deposition of raw sequencing signal, and not just the FASTQ files, is critical to the importance of this work. The deposited files still contain the fast5 raw data sequence though and should be re-basecallable with dorado, should a reader of the manuscript wish to do so.

Minor comments (these have been slightly rearranged/reordered for the response)

- *Lines 14-15: “a myriad of chemical modifications” should be “myriad chemical modifications”*
- *Figure 2D-2E and Lines 122-131 – Based on the result, I’m assuming the non-stem cells are slower/less proliferative, but it might be worth briefly mentioning the biology of the system in the text when it’s introduced*
- *Lines 134-145 – should specify “human” mitochondrial genome for clarity*
- *Figure 2J: The text of “forward” and “reverse” is too small*
- *In several places throughout the text, the authors use the language of “fraction of genomes divided”. Is there a reason the authors are using this language instead of “replicated”? (e.g. Figures 3F and 4C)*
- *Line 247: I think “the gene encoding this gene” should be “the gene encoding this protein”*

We thank the reviewer for these corrections, which we have incorporated into the text. One additional minor note – in the text describing the mitochondrial experiments, we originally referred to the transcription factor “ATFS-1”, which is actually the *C. elegans* name for a transcription factor which is

called ATF5 in humans and other mammals. Reminded by the reviewer's comment about specifying "human" genomes, we have corrected this in the revised text.

- *Lines 132-149: It's really cool that this can be a way to track synthesized DNA in vitro, and it's a nice validation of mtDNA transcription the way the cDNA aligns. For most practical purposes, I think people are going to use DNase to remove contaminating DNA from samples before doing cDNA synthesis. It might be worth discussing here or in the discussion situations where this strategy could be used.*

We apologize for the lack of clarity, but we did actually use DNase I to remove contaminating DNA from the RNA sample prior to performing cDNA synthesis in this experiment! Thank you for pointing this out - we had specifically mentioned the inclusion of the DNase I step in the methods but had omitted it from the section in the main text, and it's one of the things that makes the experiment interesting. It is entirely possible that the DNase treatment could be further optimized to remove lingering mtDNA, but the step was performed as per the instructions in the DNA isolation kit protocol so I think this is a realistic demonstration. I think the issue here is something related specifically to mtDNA – either its size or the sheer abundance of it within a cell or something similar – but either way hopefully this demonstration will be of interest to readers!

- *Figure 1F: It's not clear to me if the three shades of green/gold are denoting something different. At first I thought they aren't but they aren't the same widths. This should be clarified or simplified to just one shade each.*

Our original intent with the shading was to show that, since the library in figure 1F was sequenced across three sequencing preparations/flow cell loadings, that each preparation had similar quality benchmarks; the three shades represent the individual results from each of those batches. We agree that this is likely more confusing and distracting than beneficial, and have simplified the figure by removing the shading.

- *The text on lines 192-194 discusses that the infected conditions have less replication than the uninfected controls and refers to Figure 4C. The figure doesn't seem to distinguish between infect and uninfected human samples.*

The uninfected samples were only sequenced at 24h, and were present as the (admittedly difficult to distinguish) gray dots at the 24h position of the figure panel. We have added specific text to the panel to make these more visible.

- *Figure 4G: What are the 13% of reads that have no m6A but don't map to Hg38? I believe the text on line 223 indicates that these are likely repetitive sequences that may not align well in Hg38? Not strictly necessary for your point, but have you tried aligning to the T2T assembly instead?*

Very interesting question! Based on sequence BLASTing, they do seem to be unplaced human DNA, but faulty reads are also a likely possibility (e.g. a DNA molecule gets stuck in the pore and continually outputs a repetitive nucleotide sequence based on the sustained faulty readings caused by a non-moving nucleotide molecule). Aligning the non-hg38 reads to the T2T assembly (chm13v2), only 1.4% of those previously unmapped reads are able to be aligned with the minimap2 'map-ont' setting, but it is possible

that highly repetitive sequences may still have issues with alignment due to the mechanics of minimap2 and this may be an undercount.

Response to Reviewer #3:

Notes on David et al. 2025 Comments for the author

In this manuscript the authors introduce a novel method to produce all possible unique kmer oligos of a specific length (barcoded split-pool synthesis; BSPS) with a base modification. This is essential to support detection of modification using third generation sequencing technologies, such as ONT and Pacbio sequencing. The approach is used to determine signal levels of a specific modification in all sequence contexts and subsequently used in a number of use cases, such as screening drug sensitivity or studying mitochondrial genome dynamics. The work represents a very useful demonstration of split-pool synthesis to the detection of DNA modifications using 3rd gen sequencing platforms, which is an important advance of the field that undoubtedly will find great applications. However, given its presentation as a method there is limited attention to the method development and validation, in lieu of practical demonstration.

We thank the reviewer for this comment, and in response we would like to highlight how we view this study. The framing of the manuscript to focus on the applications rather than the method development and validation was an intentional choice. The original version of the manuscript, which we circulated to collaborators and colleagues, focused on the different types of split-pool syntheses possible and the effects on current of various types of modified bases (Extended data figures E2/E3 were originally main figures 2 and 3, for instance). The universal consensus was that the manuscript was impenetrable. Colleagues who we now have strong collaborations with, using this technology to query their systems, couldn't see past the dense technical aspects to understand how modified base sequencing could be applied to their work. That version of the manuscript would only have been of interest to scientists already exploring modified base sequencing, which we think is a very interesting field doing incredible work, but it's not a large one. We made the conscious decision to reframe the manuscript for a general audience (and submit to a broad focus journal in *Nature Communications*, rather than a more technical sequencing-focused forum) – the focus now is on demonstrating how this approach can be used across the biological sciences to enable interrogations of systems that previously would not have been interrogated with any sequencing approach, or were maybe even not possible to study. I strongly believe that this version of the manuscript will have a wider and more dramatic impact in more areas of science than a highly technical methods description would, and I do think we've included enough technical details in the associated files with the manuscript that readers interested in those aspects will be able to replicate and use our methods.

Major:

- The authors extensively use Nanopolish. Nanopolish eventalign gives “the optimal alignment” between the measured raw current signal and the expected segment values, where the latter is taken from a

lookup table that has a value assigned to all possible k-mers. Aligning the raw signal (with a modification) to this expected sequence (without any corrections done for modifications) may lead to the aligner producing a result where the values for the modified base are assigned to the neighbouring base (/overlapping k-mer) instead. Hence, looking for differences in value distributions can be difficult as the actual signal change is now mis-aligned. Nanopolish itself gets around this issue when calling methylation by employing an HMM in the region around the target motifs, hence effectively allowing the HMM to find a better local realignment solution if needed, where the alignment with both a modified and unmodified alternative of the same sequence are tested and the most likely option to have produced the measured current is called. The approach employed by the authors seems more to the approach employed in Tombo (which by the way is also not optimal).

- In relation to the point above: why did the authors not explore one of many deep learning solutions? They themselves call their data 'training libraries', so why not leverage RNN-based models which get around the issue of relying on alignment only?

I want to clarify that we see two major advances presented in this manuscript: the first is the barcoded split-pool synthesis approach that allows for the generation of the modified base k-mers, and the second is the conceptual framework of utilizing modified base sequencing to understand biology in systems that are often only tangentially related to nucleic acids. The reviewer makes great points about advanced computational approaches to enable even more complex interrogation of advanced bases – I think this is an excellent point, but it's one that is beyond the scope of the current manuscript. How we view the advances and how we've presented them in this manuscript is discussed further below, in response to another of Reviewer 3's points.

Additionally, I think our data suggest that one of the main limitations of modified base calling, at least in the R9 flow cell, is not the computational processing of the data, but the actual current output of the pore. In response to the reviewer's comments, we have added an RNN to the manuscript. The output of the RNN, our simple approach calculating z-scores, and DNAscent agree on the classification of 99.5% of reads from real data. We have also added example raw current distributions, derived from our BPS data, to Extended Data Figure E5C, comparing the results of DNAscent on BPS oligos. Our intention with the original version of this figure was not to show a problem with DNAscent, but to demonstrate that any computational approach will need to make conceptual tradeoffs based on the actual current differences induced by the base modification in the pore. With the way DNAscent was trained and programmed, it is not as sensitive at detecting rare and highly localized stretches of several base substitutions (as are found in the BPS oligos), but is more specific in detecting the start/end of broad stretches of incorporations, as it was ultimately designed to use that type of data as input for fork detection in replicating DNA. Especially given the profiles of current differences, I don't think the limitation for single-base detection is the computational approach (as seen by the highly consistent results between DNAscent (HMM-based), our simple z-score calculations, and the new RNN). The presentation of our method and the data generated by it should allow for each user to decide the tradeoffs they're willing to accept for each application of modified base sequencing (as was the case with the design of DNAscent) – in addition to the full protocol for our method now being included with the

manuscript, all of our raw sequencing data is available through SRA and is thus available for the generation of more advanced computational methods.

- For the R9 pore model it is generally assumed that the signal is influenced by 5 or 6 bases in the pore at any one time, hence 5-mers and 6-mers seem a fitting k-mer length. However, if a 5-mer influences the signal, calling a modification from just the 5-mer where the target base is in the very center ignores information that can be obtained from up to two bases to the left and right. This becomes especially important for pores that have a wider read head, such as R10. Have the authors considered this in their analysis, and might this improve the results?

As discussed in the response to Reviewer 1, for our simple analysis, the information gained from considering the additional positions is less than the noise introduced by those positions, for BrdU at least (dU seems to induce a strong effect at position 4 as well), and we did not see an increase in detection efficiency when adding them to our approach. That being said, the newly-added RNN (which does take into account this information) does seem to benefit from including that data, so it does seem like more advanced methods capable of utilizing this information will return better single-base accuracy.

- I think the estimation of the performance of the approach should be extended and not hidden in an external figure. Moreover, the 'per read' performances obscure the 'per base' performances (which are in E5). I would expect from a new method that there is an estimated error rate (at a per-base rather than per read level) of the method and that there is a comparison to the sota method on a gold standard or positive control. The suggestion of high accuracies presented in E4K and L (based on only a few 'callable kmers' with modest accuracy) seems to suggest that most signal is due to the assumption several 'callable' kmers are present in a read and not due to high accuracy of the method itself. In sum, the "statistical analysis" on the bottom of p23, deserves more attention and clarity. Of note, ROC analysis in general is not very suitable for situations where the vast majority of calls is expected to be negative, as just calling everything negative will already yield a good performance. Precision/recall/FDR estimates becomes especially relevant in case of rare modification, as in that case a non-negligible error rate can already give rise to very high false discovery rates (FDRs). It is unclear if the presented performance also holds true for the other proposed base modifications / analogs. If they can show that this BSPS provides a higher (modified) basecalling accuracy / lower FDR this would really endorse their point of the utility of this method.

We respectfully disagree with the reviewer on the framing of these points. The main point I would raise is conceptual – the innovation in our presented method is not with the computational analysis of sequencing data. As mentioned above, we see our innovations as mainly (1) the (wet bench) oligonucleotide synthesis approach that allows for the generation of these training sequences, and to a lesser extent (2) the conceptual advance of using modified base sequencing to study biological processes not necessarily directly related to those specific base modifications. The computational analysis of modified base detection is enabled by (1) and is required for (2), but our analysis approach here itself isn't a significant conceptual or computational advance in the way that, for instance, DNAscent or Nanopolish were. An important point here is that our computational analysis is roughly fourteen lines of

R commands (which are all base commands, with no packages required), yet is almost exactly as accurate as DNAscent (an HMM-based approach) – our reference data generated by the BPS protocol are strong enough that even a simple analysis can return highly accurate results. We have placed these data in the extended data because they're essential to understand what was done, but they're not the core findings of the work. (Also, I don't think it's fair to say they're "hidden" in the extended data – the editor can correct me if I'm wrong, but my understanding is that at *Nature Communications*, extended data figures are incorporated into the html version of the article on the web and appended to the PDF when downloaded; these aren't supplemental data only available to those who click on a specific link at the end of the website.)

With that said, I think this comment misses the point of showing the data in extended data figures E4-E6. We're not trying to characterize the accuracy of our computational base detection method with figure E4. The point is to demonstrate that our data make it possible to get closer to analyzing the base-detection capability of the nanopore sequencer itself. What our wet-bench synthesis protocol allows that wasn't previously possible is the generation of ground truth sequencing data. There are absolutely challenges with revealing this ground truth, as the reviewer points out with the limitations on alignment of tools like Tombo and Nanopolish; but contained within the fast5 files (that we've deposited into SRA thus making them available for any reader to analyze) are the raw current values from nanopore sequencing of each modified base in every 5-mer context – this data, and other data generated by our wet-bench approach, should greatly enhance the development of the next generation of computational tools. Additionally, the point of the read-level analysis in Ext Data Figure E4 is to show that even with a relatively poor single-base detection capability (panels E4A-C), the assignment of broad stretches of sequence (read- or region-level) is quite simple and robust (panels E4D-N). The point here is to show how the data generated by BPS allows for the design of an analysis tool/approach; we can have confidence that even our quite basic base modification detection calculations are returning highly accurate results for the experiments presented in the main figures.

The reviewer's point about the statistical detection of rare bases is a fair point, however for these bases there is no gold standard to compare to. For rarer bases, an approach like DNAscent's, using a hidden Markov model on two sets of training data, would not be possible – the rare base could not be incorporated at a high enough frequency for the HMM to detect transitions, and as mentioned, the data in Ext. Data Fig E5A/B show that DNAscent was not designed to be optimized for the detection of single rare bases. But a specific point of Extended Data E4 isn't just the demonstration of BrdU sensitivity, it's a demonstration that BPS data can be used to determine the feasibility of a modified base detection approach. Extended data figure E6 shows the single-base detection ability of the other modified bases in the study based on ROC, but if a reader wishes to program an approach to detect a rarer base, then they can use the sequencing data from a BPS-constructed approach to determine the statistical sensitivity of the nanopore and calculate the relevant performance of the nanopore in detecting that base.

- The authors rightfully point out that this approach could be the basis for basecalling methods for other pore designs, presumably hinting at the more commonly used R10 pore, which has a much longer read head. This will require many more split pool 'cycles'. It would be good to be more concrete about the

feasibility of such approach and how much coverage would be expected when kmers of e.g. 40 bp are required for model training.

This is an excellent point, and we have added an extended data figure (E10) to show potential updates to the method for future analysis. One of the advantages of our approach is its flexibility – it is possible to simply run the published protocol for nine rounds to generate the 9-mers currently utilized by the R10 pore. The math gets harder (there are $5^9 \approx 2$ million 9-mers and even minor inefficiencies in ligation compound with each round), but the relatively low cost of BSPS synthesis could permit those libraries to be synthesized and sequenced. Additionally, rounds could get ‘wider’ instead of just ‘deeper’ – instead of using five splits for nine rounds to synthesize R10 9-mers, we could also create donors that add two bases each and run twenty-five splits for the same five rounds to generate 10-mers (see Ext Data Fig E10A).

With the R10 specifically however, our initial analysis suggests that, likely since the R10 specifically consists of two pores that the DNA passes through sequentially, the R10 9-mers can be analyzed as two overlapping 5-mers, in a way that the R9 cannot simply be analyzed as two overlapping 3-mers. Figure 1J highlights that sometimes the bases in an R9 5-mer can interact in complex ways; this does not appear to be the case across the entire R10 9-mer. There are complex interactions within each pore – within positions 1-5 or 5-9, bases can interact in odd ways – but it does not appear that bases in two separate pores interact in complex ways. We are currently investigating the analysis of R10 k-mers as two distinct 5-mers, as well as synthesizing wider pools as described above, in order to generate libraries for the R10 pore, but a full description of this work is beyond the scope of the current manuscript.

We do absolutely agree with the reviewer that it is very unlikely that any adaptation to our library construction protocol would be able to generate sufficient coverage of the over 9 octillion ($5^{40} \approx 9 \times 10^{27}$) five base 40-mers. If the R11 pore design uses 40-mers, then barcoded split-pool synthesis will not be a viable method to generate the reference libraries for those pores (which, by my quick math, would each consist of several hundred tons of DNA for a library containing just 1X coverage of each 40-mer, even with no adapter or barcode sequences).

Minor

Pg21 l45: “the manufacturer’s protocol starting from the ‘Adapter ligation and clean-up’ step” needs a version or something else much more specific (perhaps written elsewhere?)

This is a very fair comment. In response to similar concerns from additional reviewers, we have included the full protocol for synthesis, sequencing, and analysis of the libraries as supplemental data, allowing any user to adapt and utilize our method. We’ve also clarified the rationale for skipping the initial nanopore kit protocol steps (our oligos are already end repaired and A-tailed due to the synthesis design, so those steps in the current and possible future protocols are unnecessary), allowing for easy adaptation to future kit protocols, rather than relying on a specific published version of the protocol.

Pg22 l7: “Reads that fully mapped”, is that reads that just map from start to end, without any hard or softclipping (rare)? Or just reads that did not get split mapped?

We apologize for the lack of clarity - these are reads for which the entire reference sequence is represented from start to end in the read, with no gaps that could indicate barcode exclusion or mismatch. This text has been clarified to address this confusion and the samtools view parameters to isolate these reads are provided in the text.

Pg22 l12-l21: I don't see how this could be done without any code.

These analysis were performed with relatively basic R commands, which we have now included examples of at the end of the included protocol.

Pg22 l20-l21: This noise correction is not explained

We have made the noise correction explicit in the manuscript and the included example code. Briefly, we also look at current variance at k-mers which statistically show identical current values in our BSPS reference data (these are usually k-mers that do not include a potential modification site – e.g. for BrdU, these are mostly k-mers without a T), under the assumption that read-level variance at these k-mers represents a noisy read rather than modified base current effects.

Response to Reviewer #4:

David et al. introduced Barcoded Split-Pool Synthesis (BSPS), which is a rapid and inexpensive experimental technique to produce reference standard, modification-containing DNA oligos. The BSPS technique is uniquely suitable for producing training data for nanopore sequencing-based modification detection algorithms. Leveraging BSPS, the authors studied various modifications, e.g. BrdU, under a broad spectrum of biological processes, e.g. DNA replication.

Producing ground-truth training data is of crucial importance for nanopore sequencing-based modification detection, and is the latest focus of the nanopore community. I therefore consider this paper to be highly significant. My comments are:

We greatly appreciate the reviewer's comments regarding the significance of our work.

1. Preventing self-ligation is essential, and the "even-odd" design is smart. Therefore, I would suggest the authors to add a schematic panel in Figure 1A to highlight the design.

This is an excellent suggestion which we have incorporated into the manuscript (although, we added it to Figure 1C for space/simplicity concerns).

2. If I understand correctly, added bases correspond to barcodes. However, in Data S1, only 7 barcodes were provided, without noting their corresponding bases. Would the authors please clarify?

Since each library was sequenced independently (on new flow cells for each library), barcodes were reused for each experiment – in general barcodes 1, 2, 3, and 4 were used for unmodified dA, dC, dG, and dT respectively, while 5 through 7 were used for the modified bases in each synthesis library as

necessary. In the BrdU library, BrdU was bc5, while in the dI library, dI was represented with bc5, etc. Barcodes 6 and 7 were only used for the CpG, mCpG, and hmCpG experiment, which were represented by bc5, bc6, and bc7, respectively. We have included the barcoding strategy for each synthesis library in new **Extended Data Table E2**, which will also assist users in analyzing our deposited sequencing data.

3. Also for Data S1, if I understand correctly, “r1” are round 1 donor reverse sequences. “N” in “n” donor reverse sequences will basepair with the modification added in the previous round. Would the authors please clarify? Meanwhile, “terminators” were not cited in the manuscript: will “V3-5term_hp_odd” and “V3-5term_hp_even” be added to the barcode side, while the “V3-3term_hairpin” to the base side when BSPS finishes? Are they also added through ligation? If so, why there are no “F” and “R” sequences?

We apologize for the lack of clarity, in the revised version of this manuscript we have included the full protocol which should hopefully provide clarity. To summarize: this is exactly correct, the r1 donors are used in round one when the overhang is constant (dA) based on the acceptor sequence – in subsequent rounds the overhang is variable based on the previous round so each possibility is included in the donor sequence (see **extended data figure E1C** for a schematic). The terminators are each hairpins (which is what the “hp” in the oligo name stands for), so the R sequence is simply the second half of the oligo as it anneals to itself. These are ligated on at the end of the synthesis, and then exonuclease digestion is used to purify the library down to remove incompletely synthesized oligos (which are not protected by these hairpins), then the 5’ terminator hairpin is cleaved to reveal a dA overhang for sequencing adaptor ligation, which also ensures that we are only sequencing the forward strands of the oligos (since due to the design of the donors, the reverse strands do not contain any modified bases).

4. Considering the above complications, I would suggest the authors provide the .fa file for “all possible fully-synthesized oligos”, or a script for enumerating such combinations, to make the BSPS bioinformatics user-friendly.

This is a great point – we have included the R script used to generate these references for each base so that any user can modify this script to create appropriate full reference fastas for each experiment.

5. I am wondering how efficient the BSPS ligation is. Maybe the authors can quantify the oligo length distribution using nanopore sequencing reads, as the quality control for BSPS?

We apologize for the lack of clarity, but this was the intention of figure 1F (and ext data E2B), showing that the full ligation efficiency is between 10% and 25% efficient.

6. Typically, R9.4.1 DNA nanopore sequencing signal events are corresponded to 6mers. For example, nanopolish (an essential bioinformatic component in this study) uses 6mer model to perform “eventalign”. I totally acknowledge that BSPS can produce 6mer training data, but could the authors please clarify the rationale behind producing 5mers?

By our initial experiments, we found that the slight increase in accuracy from 6-mer analysis was less beneficial than the increased coverage from 5-mer analysis, which was due to a combination of two less ligations and the decreased possibility space (there are $5^5 = 3,125$ 5-mers and $5^6 = 16,525$ 6-mers). As acknowledged, the protocol could easily be run for 6-mers and just sequenced more, but, our calculations (and finances) led us to synthesize and sequence 5-mers for the current work.

7. Related to the previous comment, from a technical perspective, could the authors please describe how the 5mer event tables were produced, as nanopolish eventalign produces 6mer event tables by default?

Nanopolish can be compiled to analyze 5-mers; older versions analyzed 5-mers as default and this functionality has been retained in newer versions. The k-mer size is set as a variable ($k = 5$ or 6 , with 6 as default) in “nanopolish_squiggle_read.cpp”.

8. DNA regions containing (particularly dense, such as BSPS oligos with 5 consecutive BrdUs) modifications might completely disrupt basecalling therefore producing “unmappable reads”. As a result, “characteristic modification signals” will not be recorded in nanopolish eventalign event tables: only signals fall within “canonical signal distributions” will be recorded. Did the authors observe decreased aligned read% for densely modified oligos?

This was a concern of ours and we attempted to address it with the analysis presented in Ext. Data fig E1D and similar analyses performed for other bases. From what we can see, there may be a slight effect on most simple bases, where we see slightly less coverage with increased base modifications per 5-mer, but it shouldn't be significant enough to affect the current analysis. (We are also unable to determine whether this slight decrease is due to decreased efficiency in analysis or actual BSPS oligo synthesis.) There is a very large effect on the basecalling of abasic sites, which are always miscalled, but it seems as though minimap2 is tolerant of the base miscalls as even these values can be aligned to the reference and produce extractable current values from eventalign (which calls their sequence universally as “NNNNN”). Given that even the abasic currents can produce useful values, we think that the effect on smaller current differences from less drastic base modifications are unlikely to significantly affect the results of our analysis.

Another important point is that all of the raw current data for each of our synthesized libraries is available in SRA (with the accession listed in the paper) – a reader of the manuscript with a better idea on how to analyze these data to this concern could easily reanalyze our data and generate their own current values.

9. The R9.4.1 chemistry has been discontinued and the R10.4.1 chemistry is the state-of-the-art. Therefore, discussing the compatibility between BSPS and R10.4.1 may further expand the impact of this paper.

10. RNA modification detection emerges as the state-of-the-art research focus of the nanopore sequencing community. Therefore, discussing the possibility of producing RNA oligos may further expand the impact of BSPPS.

These are excellent points and we have added an extended data figure (E10) detailing our ideas to expand this technology for the R10 technology. The conversion of the technology to RNA is non-trivial, but we are currently investigating expansions of the method in ways compatible with RNA (including the use of splint ligations and RNA-specific enzyme steps), but these approaches are not quite mature enough to include in the manuscript – we have added a line discussing the possibility of RNA expansion to the discussion.

11. No sure what does "two/multi-dimensional sequencing" mean. Maybe the authors referred to multi-omics (sequence + modification)? Maybe consider using "native DNA sequencing"?

Our intent was to highlight that while NGS is single-dimensional sequencing, producing one dimension of information (base sequencing), our approach allows for multiple dimensions of information which have been encoded into the reads – e.g. we get base sequence plus replication and source organism information (for the data presented in figure 4). Our intent is to highlight the conceptual framework of using sequencing to measure additional variables than just DNA base sequence.

12. For the METHODS section "Statistical analysis of modified base detection", it is important to clarify the number of training samples generated for each of the 2101 5mers. Specifically, it would be useful to know whether the training samples are evenly distributed across all the 5mers or if there is any data imbalance.

I apologize for the lack of clarity – the coverage per 5-mer data are presented in the supplemental data detailing the current values – Supplemental Data S3 for BrdU. A reference to this table has been added to the appropriate section to make this clearer.

Reviewer #5:

As a minor aside, I think this is a fantastic initiative that my own trainees have benefitted from, and I am glad to see this note.

Minor other corrections:

The original manuscript incorrectly referred to the transcription factor involved in mitochondrial stress response as “ATFS-1” – which is actually the *C. elegans* name for the protein. The revised manuscript has been corrected to refer to this transcription factor by its human/mammalian name, ATF5.

Upon collecting the data for resubmission, we noticed a typo in Figure 1G; the actual analyzed oligos were “CCGAC” and “CCIAC”, while the original version of the figure said “CCGAG” and “CCIAG”; this has been corrected for resubmission.

RESPONSE TO REVIEWERS' COMMENTS

Reviewer #1 (Remarks to the Author):

The authors have satisfactorily addressed all of my comments. I expect this paper will have a strong impact and will spur many others in this research field.

We thank the reviewer for the constructive comments and their help in refining the manuscript to create a hopefully useful resource for the field.

Reviewer #2 (Remarks to the Author):

In the revised version of the manuscript, the authors have addressed my major concerns. The addition of a detailed supplementary protocol (and numerous scripts) for both BPS library synthesis and data analysis, which should make this approach much more accessible to potential users. The authors also trained an RNN model on their BrdU data and incorporated this in their comparison of BrdU detection with DNAscent. Outside of these significant changes, there were numerous smaller tweaks that improve clarity throughout, as well as the addition of an additional extended figure detailing potential expansions of the approach. I had one minor comment, which is that Extended Data Figure E6A, the image quality of the added matrices is too low and they aren't legible (it looks good in the Python notebook).

We thank the reviewer for the constructive comments and their help in refining the manuscript, and for the comment about the matrices, which have been recopied from the source to increase the resolution in the final figure.

Reviewer #2 (Remarks on code availability):

Scripts are organized and divided between a Python notebook overviewing RNN training and scripts for processing ONT sequencing data. I was not able to test all of the scripts, but they all seem relatively straightforward and should be easy to pick up for someone with some basic bioinformatic and ONT pipeline familiarity.

As someone with not much more than basic bioinformatic familiarity, I agree with the reviewer and thank them for checking some of the scripts.

Reviewer #4 (Remarks to the Author):

The authors have addressed all my questions.

Reviewer #4 (Remarks on code availability):

NA

We thank the reviewer for their constructive comments and help in refining the manuscript.

Reviewer #5 (Remarks to the Author):

We thank the reviewer for their constructive comments and hope our manuscript was useful training in peer review.